# Brief communication: The role of using precipitation or river discharge data when assessing global coastal compound flooding

Emanuele Bevacqua[1], Michalis I. Vousdoukas[2], Theodore G. Shepherd[1], and Mathieu Vrac[3]

[1]Department of Meteorology, University of Reading, Reading, United Kingdom
[2]European Commission, Joint Research Centre (JRC), Ispra, Italy.
[3]Laboratoire des Sciences du Climat et de l'Environnement, CNRS/IPSL, Gif-sur-Yvette, France.

**Correspondence:** Emanuele Bevacqua (e.bevacqua@reading.ac.uk)

**Abstract.** Interacting storm surges and high water-runoff can cause compound flooding (CF) in low-lying coasts and river estuaries. The large-scale CF hazard has been typically studied using proxies such as the concurrence of storm surge extremes either with precipitation or with river discharge extremes. Here the impact of the choice of such proxies is addressed employing state-of-the-art global datasets. Although being proxies of diverse physical mechanisms, we find that the two approaches show similar CF spatial patterns. On average, deviations are smaller in regions where assessing the actual CF is more relevant, i.e. where the CF potential is high. Differences between the two assessments increase with the catchment size and our findings indicate that CF in long rivers (catchment $\gtrsim$ 5-10,000 $Km^2$) should be analysed using river discharge data. The precipitation-based assessment allows for considering local rainfall-driven CF, and CF in small rivers not resolved by large-scale datasets.

## 1 Introduction

Compound flooding (CF) happens in low-lying coastal areas due to the interaction of high precipitation runoff and high sea level. The combination of the two hazards can cause larger damages than those caused by the either of hazards in isolation, and recent events have occurred in, e.g., Mozambique (2019), Texas (US, 2017), the Shoalhaven Estuary (Australia, 2016), Ravenna (Italy, 2015), Cork (Ireland, 2009), and Lymington (United Kingdom, 1999) (Couasnon et al., 2020; Zscheischler et al., 2018; Kumbier et al., 2018; Bevacqua et al., 2017; Olbert et al., 2017; Hendry et al., 2019). Practitioners and the scientific community are becoming more aware of CF risk, and there were several recent studies addressing the phenomenon at local (Bevacqua et al., 2017; Kumbier et al., 2018; van den Hurk et al., 2015) or larger scales (Wahl et al., 2015; Bevacqua et al., 2019; Couasnon et al., 2020; Ward et al., 2018; Ganguli and Merz, 2019a, b; Wu and Leonard, 2019; Wu et al., 2018). Recent advances in large-scale sea level and river discharge modelling allowed the generation of sub-daily time-series of water levels along the global coastline (Vousdoukas et al., 2018; Muis et al., 2016), thus enabling continental CF assessments (Ward et al., 2018; Bevacqua et al., 2019; Couasnon et al., 2020).

CF can be the result of different mechanisms depending on the local topography and meteorology. According to Wahl et al. (2015), CF is possible to occur when: (1) the joint occurrence of high river discharge and storm surge in estuarine regions may elevate water levels to a point where flooding is initiated or its impacts exacerbated; (2) a destructive storm surge, which already caused widespread flooding, is followed by rainfall, as the latter can drive additional flooding, even if it is not an extreme event

on its own; and (3) a moderate storm surge occurs which does not directly cause flooding, but is high enough to fully block or slow down gravity-fed storm water drainage, and as a result precipitation causes flooding. In addition, CF may occur if (4) precipitation falls on wet soil that is saturated by a preceding storm surge (Bevacqua et al., 2019).

Quantifying the actual CF, i.e. the water level resulting from the combination of pluvial/fluvial flooding and high sea level, is challenging. For example, quantifying and interpreting CF in the vicinity of rivers (mechanism 1) requires water level measurements in the river mouth, which are rare, probably because most gauges are installed to monitor either riverine or marine processes (Bevacqua et al., 2017; Paprotny et al., 2018). Model-based data are also limited because only recently have modellers started considering CF. Statistical and hydrodynamic modelling approaches integrating fluvial and sea-level flooding have been developed and applied recently at the local scale (Bevacqua et al., 2017; van den Hurk et al., 2015; Kumbier et al., 2018; Khanal et al., 2019); however, at large scale, these approaches are only now being developed. Similarly, explicit study of the actual CF water level due to pluvial flooding and storm surges (mechanisms 2-4) has not received much attention so far, to our knowledge, which may also be due to the scarcity of data. Thus, to gain information on the CF hazard at the regional, continental, or global scales, scientists usually employ proxies of flood hazard, e.g. the probability of potential CF (Bevacqua et al., 2019; Ward et al., 2018; Wahl et al., 2015; Couasnon et al., 2020; Paprotny et al., 2018). Quantifying potential CF, under the present or future climate, is useful as it allows identifying potential hotspots of CF hazard. Then, more detailed assessments of the local CF risk can be carried out at such hotspots, using computationally intensive methodologies that integrate all the hydrological and meteorological sources of flooding and their physical interaction (Wahl et al., 2015).

The large-scale assessment of potential CF includes the analysis of the probability (or return periods) of concurring extreme values of CF drivers. Two main approaches exist, focussing - respectively - on the analysis of the probability of concurring high values of sea level and river discharge, or of sea level and precipitation. Through considering river discharge, the first approach takes into account CF in estuaries and deltas, i.e. serving as a proxy of CF mechanism 1. The second approach, based on the analysis of accumulated precipitation around the coast when the high sea levels occur, can represent CF due to local precipitation extremes, i.e. related to mechanisms 2, 3, and 4. Given that precipitation is among the main drivers of river discharge, the two proxies are correlated to a certain extent and the use of precipitation can thereby allow quantifying CF potential also in certain river estuaries. However, the correlation between the two proxies can be sometimes poor, especially in locations where river discharge is strongly influenced by other factors such as snowmelt, evaporation, and accumulated precipitation over previous weeks or months (Blöschl et al., 2019).

Given the scarcity and heterogeneous distribution of in situ data (Ward et al., 2018; Couasnon et al., 2020; Wu et al., 2018), scientists have started to employ model data - of river, storm surge, and precipitation - to assess the large-scale potential CF hazard (Ward et al., 2018; Bevacqua et al., 2019; Wu et al., 2018; Couasnon et al., 2020; Wu et al., 2018; Paprotny et al., 2018; Bevacqua et al., 2020). Against the foregoing background, the present study aims to assess whether a precipitation based large-scale CF assessment can be used as a surrogate for potential CF in estuaries at the large-scale. To that end we use coherent global model datasets of storms surges (including wave effects) (Vousdoukas et al., 2018), precipitation (Beck et al., 2017b), and river discharge (Couasnon et al., 2020; Eilander, 2019) and conduct a first global comparison of the results obtained through the two approaches, keeping all the other methodological aspects identical.

## 2 Data

We analyse the period 1979-2015. We consider river discharge daily maxima from a publicly available global dataset (Eilander, 2019; Couasnon et al., 2020), which includes coastal catchments larger than 1,000 km$^2$. The dataset was the result of hydrological model simulations forced with temperature and potential evaporation derived from ERA-Interim, and with precipitation from the MSWEPv1.2 dataset (Couasnon et al., 2020). The latter is obtained by merging gauge, satellite, and reanalysis data (including ERA-Interim); more information can be found in Beck at al. (Beck et al., 2017b).

Precipitation is taken from the same MSWEPv1.2 dataset used to simulated river discharges and consists of daily data over a 0.25° grid. On each day we consider accumulated precipitation amounts within a 3-day centered window. This enables us to account for precipitation occurring just before and after midnight of the storm surge day (Bevacqua et al., 2019; Martius et al., 2016), and to consider different mechanisms causing CF (Wahl et al., 2015; Bevacqua et al., 2019).

Storm surges and waves were simulated with the hydrodynamic model DFLOW FM (Vousdoukas et al., 2017, 2018) and the wave model Wavewatch III (Mentaschi et al., 2017; Vousdoukas et al., 2017, 2018). The wave model was forced by 6-hourly wind, while DFLOW-FM was also forced by sea level pressure fields, both available from the ERA-Interim reanalysis (Dee et al., 2011). The effects of tropical cyclones (TCs) were considered in the reanalysis through storm surge simulations forced by downscaled atmospheric fields from all recorded TCs and by considering satellite-observed TC wave extremes (Vousdoukas et al., 2018). Astronomical tides are not considered in this analysis in order to focus on the meteorological component of the sea level, and excluding the stochastic coupling with tide induced water level variations. We consider daily water level maxima from the combined result of storm surges and wave setup (hereinafter mentioned as storm surges) according to Vousdoukas et al. (2018).

We analyse CF only around river mouth locations whose nearest precipitation and storm surge grid points lie within a distance of 75 km (Couasnon et al., 2020). This results in considering locations at river mouths of catchments with size in between about 1,000 and 3,690,000 Km$^2$ (95% having size smaller than 50,000 Km$^2$; Fig. 3f).

### 2.1 Methods

We assess the potential CF hazard via bivariate return periods of concurring extreme events (Vandenberghe et al., 2011; Manning et al., 2019) of the variables X and Y, i.e. storm surge and precipitation (CF$_{prec}$) or storm surge and river discharge (CF$_{river}$). Extremes of the individual variables ($x_{ext}$ and $y_{ext}$) are defined as the associated $\alpha$-year return levels. We use different $\alpha$ in the following, though we present the main results for $\alpha$=5; images for $\alpha$=2 are shown in the supplementary material. Return levels are obtained through fitting a generalized extreme value distribution to the annual maxima of the individual variables (Coles et al., 2001). Annual maxima are defined based on adjacent windows centred on the month where the climatological river discharge average is the highest, rather than from January-December windows (such a window definition reduces the chance of selecting two consecutive annual maxima belonging to the same river discharge extreme event, and therefore leads to a more robust definition of the return levels). Overall, given the definition of the extremes based on $\alpha$-year return levels, the bivariate return period is inherently linked to the dependence of the pairs in the tail of the distribution.

CF bivariate return periods are computed following the methodology presented by Bevacqua et al. (2019). The CF return period computation is based on the bivariate distribution of the variables of interest (X,Y), which is estimated semi-empirically to allow for robust estimation. For a given location, we select pairs whose individual values are simultaneously larger than the individual 95th percentiles ($x_{\mathrm{sel}}$ and $y_{\mathrm{sel}}$), and we model these pairs via a copula-based distribution. If the defined thresholds result in a small group of selected pairs, we lower the 95th percentile selection threshold to guarantee having at least 20 pairs. The choice of 20 pairs is a trade off between having a sufficient amount of selected pairs and employing a reasonably high threshold for the fit of the parametric distribution in the tail. Furthermore, the return periods are largely insensitive to changes in the threshold (results are similar based on 20, 30, and 40 pairs; not shown). The selection thresholds are generally high: 75% of the locations have a selected-threshold larger or equal to 0.95 and 0.94 for the precipitation- and river-based analysis, respectively. And 95% (99%) of the locations have a selected-threshold above 0.93 (0.885) and 0.89 (0.85) for the precipitation- and river-based analysis, respectively.

Once pairs are selected, clusters of pairs separated by less than 3 days were considered as part of the same event represented by the maximum X and Y values observed in the cluster. Note that while this choice has the drawback of not fully respecting the assumptions of independent realisations of the extreme events, which is necessary to apply extreme values theory in its generic form, it allows considering multiple storm surges that may occur during a sustained period of high river discharge and that could lead to multiple compound floods.

The return period is defined as:

$$T(x_{\mathrm{ext}}, y_{\mathrm{ext}}) = \frac{\mu}{P((x > x_{\mathrm{ext}} \ and \ y > y_{\mathrm{ext}}) \mid (x > x_{\mathrm{sel}} \ and \ y > y_{\mathrm{sel}}))} =$$
$$= \frac{\mu}{1 - u_{\mathrm{Xext}} - u_{\mathrm{Yext}} + C_{\mathrm{XY}}(u_{\mathrm{Xext}}, u_{\mathrm{Yext}})} \tag{1}$$

where $\mu$ is the average time elapsing between the selected pairs, $u_{\mathrm{Xext}} = F_{\mathrm{X}}(x_{\mathrm{ext}})$, $F_{\mathrm{X}}$ is the marginal cumulative distribution of the excesses over the selection threshold (accordingly for Y), and $C_{\mathrm{XY}}$ is the copula modelling the dependence between the selected pairs (see Fig. A1 for the dependence associated with the fitted copulas in the two assessments). Note that, as the return period is obtained as a combination of the average elapsing time $\mu$ and the parametric probability density function of the data in the tail, an exact correspondence between the dependence of the copula and the return period is not expected. We model the marginal distributions of X and Y beyond the selection thresholds by a Generalised Pareto Distribution (GPD). We fit copulas from the families Gaussian, t, Clayton, Gumbel, Frank, Joe, BB1, BB6, BB7, BB8 to $(u_X, u_Y)$ (obtained via empirical marginal cumulative distribution function (Vandenberghe et al., 2011; Manning et al., 2018)); then we select the best ranked family according to the Akaike information criterion. In general, the physical processes captured by the two assessments can differ (even at the same location), therefore we allow for the selection of different copulas in the two assessments. We fit copulas and marginal distributions via a maximum likelihood estimator (using the *VineCopula* (Schepsmeier et al., 2016) and *ismev* (Heffernan et al., 2016) R-packages). We test the goodness of fit of copulas and marginals via the Cramer-von-Mises criterion (via the *eva* (Bader and Yan, 2016) and *VineCopula* (Schepsmeier et al., 2016) R-packages respectively).

When referring to the assessment of whether the CF return periods based on river discharge ($T_{\mathrm{river}}$) are statistically different from those based on precipitation ($T_{\mathrm{prec}}$) or not, we use the concept of statistical compatibility, recently introduced by Amrhein

et al. (Amrhein et al., 2019). We compute the centered 95% (2.5-97.5%) confidence interval of $T_{prec}$ on the basis of 600 resampled bivariate time series of precipitation and storm surge (each of them built randomly combining observed 1 calendar year bivariate time series (Bevacqua et al., 2020)). $T_{river}$ is regarded as being statistically compatible with $T_{prec}$ if $T_{river}$ lies within the 95% confidence interval of $T_{prec}$, and incompatible otherwise.

We qualitatively investigate how the two assessments compare for different classes of catchment size. To do so, we rank the rivers based on their catchment size and divide them into groups having the same sample size; for each group we compute different statistics to compare the two assessments: Spearman correlation of $T_{river}$ and $T_{prec}$, ratio $T_{river}/T_{prec}$, and percentage of locations with $T_{river}$ compatible to $T_{prec}$. This binning procedure provides equally robust statistics for each bin and shows similar results for small variations in the bin size.

### 2.1.1   Results and discussion

The spatial patterns of the potential CF return periods based on either precipitation ($T_{prec}$), or river discharge ($T_{river}$) are very similar (Fig. 1; Fig. A2 is identical but shows results based on extremes defined considering 2-year return levels). The results for clusters of locations with the 5% lowest CF return periods are also similar in the two assessments (Fig. A3 and A4). These hotspot regions are mainly found along the US and central American coasts, Central Chile, Madagascar, the southern North

Atlantic coasts, and southern Japan.
    While the spatial patterns of the CF return periods obtained from the two approaches are very similar, their relative differences can be substantial, especially at certain locations (Fig. 2 and A5). Given that the return period computation procedure involves several uncertainty factors (e.g. bivariate model fitting and definition of the return levels), we test the hypothesis that the return period based on the river discharge is statistically compatible (at 95% confidence level) with that based on

precipitation. When defining extremes based on the 5-year return levels, the river-based return period is compatible with the precipitation-based value in about 82% of the locations (Fig. 2c; 76% for 2-year return levels: Fig. A5c). The spatial distribution of locations where $T_{river}$ is not statistically compatible with $T_{prec}$ does not seem to follow a clear spatial pattern, though it appears that $T_{river}$ is lower than $T_{prec}$ in northern Europe and in the tropics. The latter are areas where CF is unlikely. $T_{river}$ is higher than $T_{prec}$ along the Gulf of Mexico (Fig. 2c and Fig. A5c). Compatible $T_{river}$ and $T_{prec}$ are found but with large discrep-

ancies in the tropics and above $60^o$ North (Fig. 2b), consistently with the high uncertainty of these large CF return periods that do not allow for detecting potential differences between $T_{river}$ and $T_{prec}$.
    In areas with a tendency towards high CF return periods, e.g. the tropics, neighbour locations show divergent values in the ratio between the return periods of the two assessments (dark blue and red dot in Fig. 2). Further tests showed that this behaviour is not related to goodness of fit of the bivariate distributions, rather it appears associated with the large uncertainties of high

return periods and potentially with different catchment characteristics.
    We find that there is a tendency towards higher differences in the two assessments at locations where either or both $T_{prec}$ and $T_{river}$ are high (i.e. where T approaches the value expected under independence of the CF drivers) (Fig. A6). (This appears consistent with the high uncertainty associated with large CF return periods.) Such a finding has relevant implications, as it

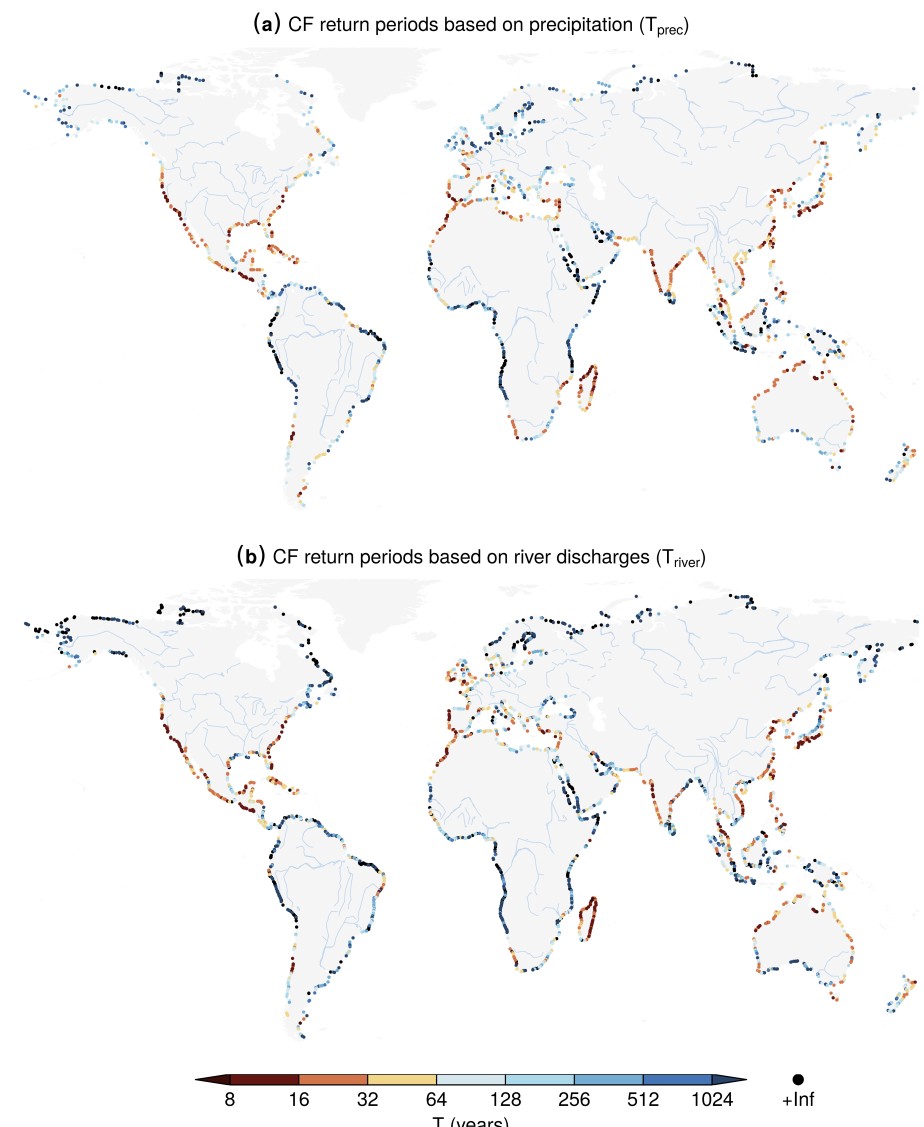

**(a)** CF return periods based on precipitation (T$_{prec}$)

**(b)** CF return periods based on river discharges (T$_{river}$)

| 8 | 16 | 32 | 64 | 128 | 256 | 512 | 1024 | +Inf |

T (years)

**Figure 1.** Present-day (1980-2014) potential compound flooding probability based on precipitation and on river discharge. Return periods of CF defined as co-occurring extremes (5-year return levels) of the CF drivers. Return period of co-occurring **(a)** storm surge (including waves) and precipitation (accumulated 3-day centred) extremes; **(b)** storm surge (including waves) and river discharge extremes. Major rivers are shown in light blue.

indicates that the two assessments tend to be similar, on average, where assessing the actual CF is more important, i.e. where there is a relatively high CF potential (Fig. A6).

The spatial association of the return periods map obtained from the two assessments is shown in Fig. 3a and A7a. The Spearman correlation between $T_{river}$ and $T_{prec}$ is ∼0.7, and increases as the return level threshold employed to define the

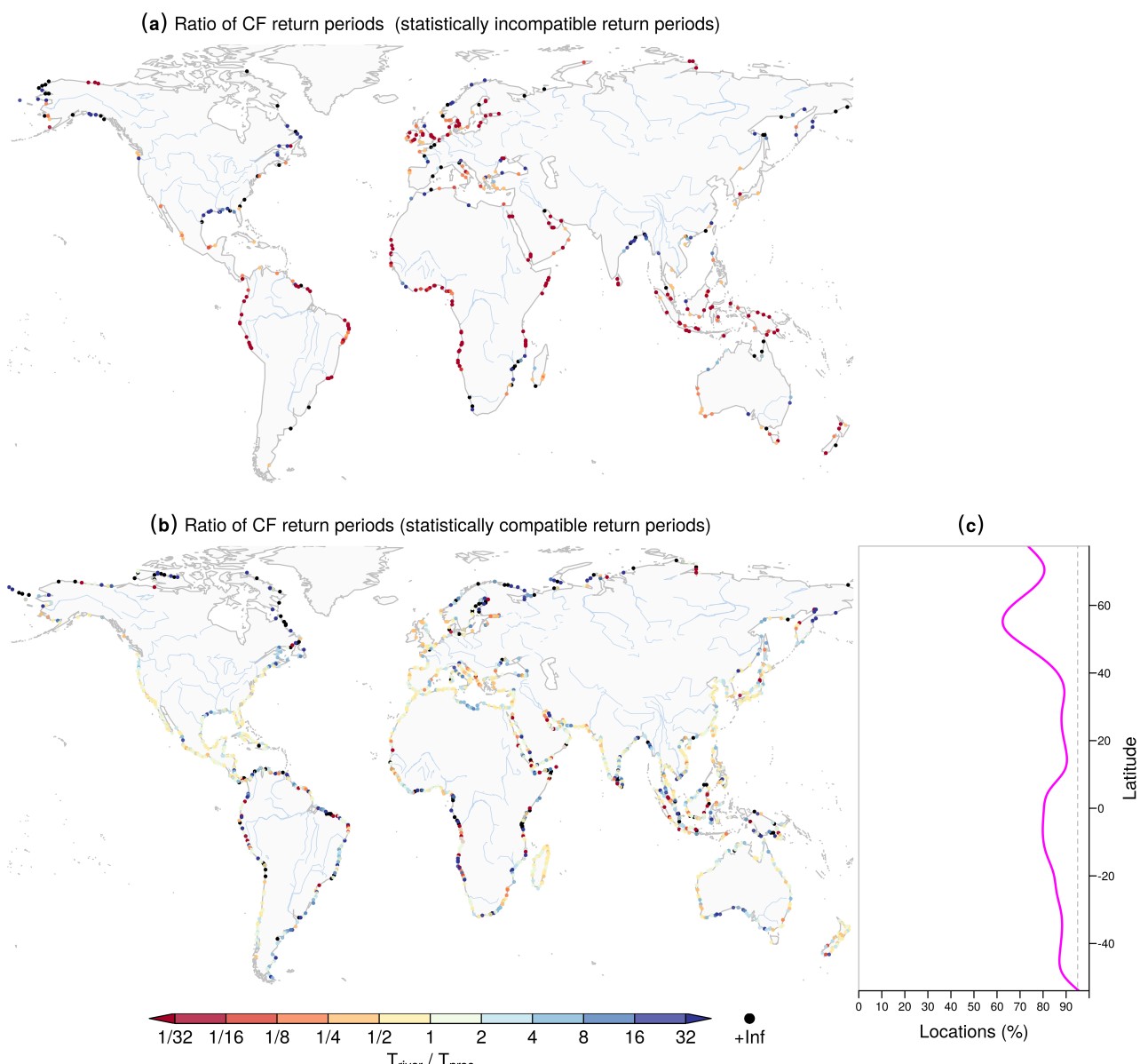

**Figure 2.** Difference between the present-day (1980-2014) potential compound flooding probability based on precipitation and on river discharge. Ratio ($T_{river}/T_{prec}$) of CF return period of concurring river discharge and storm surge extremes ($T_{river}$), to CF return period of concurring precipitation (accumulated 3-day centred) and storm surge extremes ($T_{prec}$). Panel **(a)** shows the ratio where $T_{river}$ is statistically incompatible (at 95% confidence level, see methods) with $T_{prec}$. Panel **(b)** shows the ratio where $T_{river}$ is statistically compatible with $T_{prec}$, while panel **(c)** shows the coastline fraction where this happens (binned every 5° of latitude and smoothed using a spline function). In (c), the dashed grey line shows the 95% level.

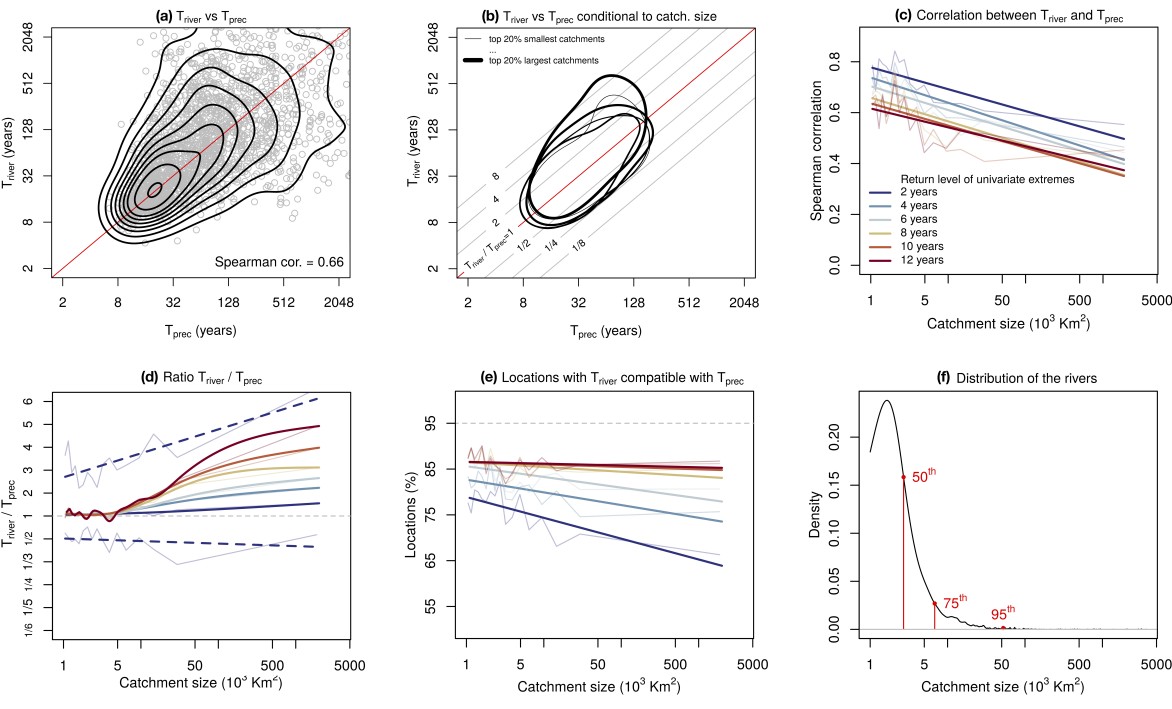

**Figure 3.** Comparison of the potential compound flooding probability based on precipitation and on river discharge. **(a)** Scatterplot (based on Fig. A2) of CF return periods based on river discharge ($T_{river}$) and on precipitation ($T_{prec}$), for extremes defined based on 5-year return levels. Black contours represent the isolines of the kernel density containing from 10% to 90% of the ($T_{river}, T_{prec}$) pairs. The identity line is shown in red. **(b)** Empirical probability density function of the variables ($T_{river}, T_{prec}$) conditioned on the catchment size; the thickness of the isolines increases with the catchment size, such that each bin considers 1/5 of the total number of analysed rivers (3178). Each contour line (isolines of the kernel density) contains about 50% of the ($T_{river}, T_{prec}$) pairs. **(c-e)** Statistics comparing $T_{river}$ and $T_{prec}$ as a function of the catchment size and of the return levels used to define the CF univariate extremes (see legend in panel (a)). **(c)** Spearman correlation between the maps of $T_{river}$ and $T_{prec}$. **(d)** $T_{river}/T_{prec}$ ratio. The dashed line is the centered 68% (16th-84th percentiles) confidence interval of the ratio based on the 2-year return levels. A non linear Y-axis for values below 1 is plotted such that the specular cases, e.g., ratio r=2 ($T_{river} = 2 \cdot T_{prec}$) and r=1/2 ($T_{prec} = 2 \cdot T_{river}$) (see Fig. 3b) appear symmetric with respect to r=1 ($T_{river} = T_{prec}$). (To obtain the plot, the ratio is defined as $r = T_{river}/T_{prec}$; then if r<1, r is transformed to $r = -T_{prec}/T_{river} + 2$. Now, e.g., r=2 and r=0 represent specular cases, therefore, where the y-axis is r=0,1,2,..., we can write r=1/2,1/3,1/4,... ). See Fig. A7b for the plot with standard axis. **(e)** Percentage of locations where $T_{river}$ is statistically compatible with $T_{prec}$ at the 95% confidence level. **(f)** Probability density function of the catchment size of the analysed rivers; 50th, 75th, and 95th percentiles of the distribution are shown in red. In **(c-e)**, tick lines are obtained through regressing the obtained statistical values per bin (thin lines) to the natural logarithm of the mean size of the catchment bins. All lines are regressed using linear regression, except from the median of the ratio in **(d)** where a spline function is used. The slopes of the linear regressions are all significant in **(c)** (p-value < 0.0024), and significant up to the 6-year return level in **(e)** (p-value < 0.022).

extremes decreases; e.g., 0.66 and 0.73 for the 5- and 2-year return level, respectively (Fig. 3a and A7a). This trend in the correlation is consistent with the higher uncertainties characterizing larger return periods.

Although some of the differences between the two assessments can be driven by uncertainties in the return period estimation, also several physical processes, e.g., topography-dependent, may be involved. The Spearman correlation between the two assessments decreases (p-value < 0.0024) as the catchment size increases (Fig. 3c). The latter suggests that - for larger catchments - different processes may cause either positive or negative deviations between $T_{\text{river}}$ and $T_{\text{prec}}$.

Water levels in the mouth of small rivers are expected to be largely influenced by the precipitation around the coast (Hendry et al., 2019; van den Hurk et al., 2015; Bevacqua et al., 2017), while around large rivers inland hydrological processes are usually dominant. Therefore, we qualitatively investigate whether the actual difference between the two assessments (quantified as $T_{\text{river}}/T_{\text{prec}}$) depends on the size of the catchment (Hendry et al., 2019). For example, the $T_{\text{river}}/T_{\text{prec}}$ ratio, defined using the 5-year return level, tends to increase with the catchment size (Fig. 3b; see the deviation of the scatter plot of $T_{\text{river}}$ vs $T_{\text{prec}}$ from the identity line emerging with the increasing dimension of the catchment). For CF return periods defined based on 2-year return level extremes the median $T_{\text{river}}/T_{\text{prec}}$ is near unity for most of the catchments, even though $T_{\text{river}}/T_{\text{prec}}$ increases slightly for larger catchments (blue line in Fig. 3d). For higher return levels, the median $T_{\text{river}}/T_{\text{prec}}$ increases with the size of the catchment (Fig. 3d), indicating that the CF assessments based on precipitation and river discharge differ largely for large rivers. For all return levels, the median $T_{\text{river}}/T_{\text{prec}}$ is near unity for rivers whose catchment size is up to about 5-10,000 Km$^2$ (Fig. 3d), i.e. about 75% of the rivers presently analysed (Fig. 3f). Binning the rivers per catchment size we find that the variance of the $T_{\text{river}}/T_{\text{prec}}$ ratio within each bin tends to be smaller for small than for large rivers (Fig. 3d, shown by the dashed line for the 2-year return level). However, it is important to highlight that there is substantial variance (Fig. 3d for the 2-year return level extremes), and that for a river of any catchment size, the associated $T_{\text{river}}/T_{\text{prec}}$ can be either small or large. Overall, the similarity of $T_{\text{river}}$ and $T_{\text{prec}}$ for small rivers is highlighted by the fact that for small rivers it is more likely that $T_{\text{river}}$ is statistically compatible with $T_{\text{prec}}$ (Fig. 3e). For example, Fig. 3e (4-year return level curve) shows that $T_{\text{river}}$ is statistically compatible with $T_{\text{prec}}$ for ∼82% of the small rivers (catchment size < 5,000 Km$^2$) and for ∼75% of large rivers (catchment size > 50,000 Km$^2$). The decrease in the compatibility of $T_{\text{river}}$ and $T_{\text{prec}}$ with the catchment size (Fig. 3e) is statistically significant for return levels smaller than the 6-year return level (p-value < 0.022). However, for high return levels this decrease is not prominent, most likely due to the large uncertainty associated with longer return periods. Fig. 3e indicates the discrepancy for large catchments in $T_{\text{river}}/T_{\text{prec}}$ being greater for low return levels, whereas Fig. 3d indicates the opposite; this is also likely caused by the larger uncertainty of return periods of higher return levels, that does not allow for detecting potential differences between large values of $T_{\text{river}}$ and $T_{\text{prec}}$.

The differences between $T_{\text{river}}$ and $T_{\text{prec}}$ are not only controlled by the catchment size, but can be the result of several other factors. During the hydrological modelling, input data artefacts, or model inaccuracies, among others, introduce uncertainty which may contribute to the observed differences. Another important contribution should arise from the uncertainties in the bivariate return period estimation (Bevacqua et al., 2019; Wahl et al., 2015), which can contribute to both positive and negative deviations between $T_{\text{river}}$ and $T_{\text{prec}}$. Moreover, the catchment response time to precipitation depends also on rock and soils catchment permeability (Hendry et al., 2019). Finally, river discharge is influenced not only by local coastal precipitation, but

also by the weather over the previous weeks or months over the catchment including evaporation and potentially snowmelt (Couasnon et al., 2020; Bevacqua et al., 2017).

Clearly, the diversity of the physical processes leading to $CF_{prec}$ and $CF_{river}$ is very relevant and can cause both positive and negative differences between $T_{river}$ and $T_{prec}$ (Blöschl et al., 2019). For example, for any given catchment, a slow catchment response time may either increase or decrease the $T_{river}/T_{prec}$ ratio. In locations where cyclones cause frequent concurring storm surge and widespread coastal precipitation, it is not guaranteed that $CF_{river}$ probability will be also high. For example, if the rainfall in the catchment upstream needs time to reach the coast, long enough for the storm surge to recede, then $CF_{river}$ will be unlikely and in this case the $T_{river}/T_{prec}$ ratio will be high (Klerk et al., 2015; Ward et al., 2018). In contrast, where the $CF_{prec}$ is unlikely because different weather systems cause precipitation and storm surge extremes, a relatively slow catchment response time may sometimes allow for high river discharge and storm surge to concur if, e.g., a first cyclone causes precipitation driving high river discharge reaching the coast when a second cyclone drives a storm surge (contributing to low $T_{river}/T_{prec}$) (Bevacqua et al., 2019).

In addition the presence of a pronounced annual cycle in river discharge can modulate the river discharge driven CF hazard, and thus the $T_{river}/T_{prec}$ ratio. In regions where $CF_{prec}$ is likely, $CF_{river}$ may be unlikely if the storm surge season does not coincide with the season of the high river discharge (Ward et al., 2018) (high $T_{river}/T_{prec}$). In contrast, where the $CF_{prec}$ is unlikely, $CF_{river}$ may be more likely if the storm surge season coincides with the high river discharge season (low $T_{river}/T_{prec}$). In addition, precipitation extremes are typically short in duration, while river discharge extremes are less dynamic. Although events exceeding the $\alpha$-year return level threshold occur both for precipitation and river discharge on average every $\alpha$ years, the number of days with high river discharge can be larger than the number of days with high precipitation amount. From a statistical point of view, the above mechanism alone would make $CF_{river}$ more likely than $CF_{prec}$ (low $T_{river}/T_{prec}$). (This effect is even more pronounced in catchments with a slow response time and especially in areas where different weather systems cause precipitation and storm surge extremes (Bevacqua et al., 2020).) The above effect is weakened as the return level threshold used to define extremes increases; due to the shorter duration of river discharge extremes, the above considerations justify $T_{river}/T_{prec}$ increasing with the return level choice (Fig. 3d).

While the relevance of these mechanisms may depend on the local climate, they are expected to be less relevant in very small rivers where the catchment response time is small, and thus autocorrelation in the river time series is smaller. Consistently, we find that $T_{river}$ and $T_{prec}$ tend to match more in smaller catchments. However, a higher agreement for small rivers might also arise from the relatively coarse spatial resolution of the data which would attenuate differences between precipitation and river discharges in small rivers. Overall, we find that independent of the catchment size, $T_{river}$ tends to be higher than $T_{prec}$ on average (Fig. 3d), suggesting that the mechanisms causing $T_{river} > T_{prec}$ may be more likely or relevant. Apparently this is a general remark and not a universal law since there are also several locations where $T_{prec}$ exceeds $T_{river}$.

The presented results are based on state-of-the-art model data which have been validated and discussed in previous papers (Vousdoukas et al., 2018; Beck et al., 2017a; Couasnon et al., 2020; Bevacqua et al., 2019; Muis et al., 2016). However, the above mentioned validation efforts did not include areas where field measurements are scarce, like parts of South America, Africa, and Asia (Ward et al., 2018). At high latitudes, ice and snow dynamics apply a certain control to both river hydrology

(Yamazaki et al., 2014) and wave and storm surge dynamics (Vousdoukas et al., 2017). However, such processes are not resolved by the global models used to generate the forcing datasets used in the present study. For these reasons, the present findings should be interpreted with care, especially in northern regions (Couasnon et al., 2020).

The two approaches investigated here provide information only on the *potential* for CF. The actual CF occurrence depends also on the local topography which can favour or not the interaction between the CF drivers; also, concurrent but not hydrologically-interacting storm surge and pluvial or fluvial flooding are relevant as they can, e.g., limit the ability to respond to emergency, and amplify the impacts that the two hazards would have caused if they occurred in isolation (Martius et al., 2016; Barton et al., 2016; Zscheischler et al., 2019). Moreover, while the two approaches investigated here are supposed to represent different CF mechanisms, separating the CF mechanisms in this way could be misleading, as CF may happen due to a combination of river discharge, local rainfall and associated surface runoff, together with high sea level. For example, in July 2019, New Orleans (Louisiana, US) (Vahedifard et al., 2016) was simultaneously threatened by the tropical Storm Barry causing local rainfall and storm surge around the coast, and by extremely high water discharge from the Mississippi River which lasted from March to July. Local CF risk assessment at the local scale should be carried out via complex hydrological modelling that can take into account the complex mechanisms causing CF, including storm surges, waves, astronomical tides, and when necessary not only fluvial or pluvial flooding but also their combination.

## 3 Conclusions

We conduct two global-scale potential CF hazard assessments, using either storm surge and precipitation, or storm surge and river discharge model data, and compare how the choice of precipitation versus river discharge as covariate with storm surge affects the results. We find that the two approaches result in similar spatial CF hazard patterns, which tend to deviate as the river catchment size increases. In addition, on average the deviations between the two assessments are smaller in regions where assessing the actual CF is more relevant, i.e. where the CF potential is high.

Due to data scarcity, current large-scale CF assessments rely on approaches and model-based datasets similar to those used here. This study indicates that for these large-scale assessments, a precipitation-based CF analysis can provide satisfactory information on the CF potential in estuaries of small and medium size rivers (catchment smaller than about 5-10,000 $Km^2$). Moreover, a precipitation-based CF analysis allows for assessing the CF hazard arising from the interaction of local coastal rainfall and storm surges where no rivers exist, or along the mouths of small rivers often not represented in global river datasets. Naturally, employing river discharge data should always be preferred to using precipitation when studying both the large and local scale CF in estuaries, when data are available, especially in areas where we detected large differences between the two approaches. The importance of using river discharge data is even greater in estuaries of long rivers.

*Data availability.* Precipitation data is available on request online (https://ec.europa.eu/jrc/en/publication/mswep-3-hourly-025-global-gri dded-precipitation-1979-2014-merging-gauge-satellite-and-reanalysis). Sea level data are available at https://data.jrc.ec.europa.eu/collecti

on/liscoast (further inquiries should be addressed to MIV). River discharge data were obtained from the dataset "Paired time series of daily
discharge and storm surge" are available at: https://doi.org/10.5281/zenodo.3258007.

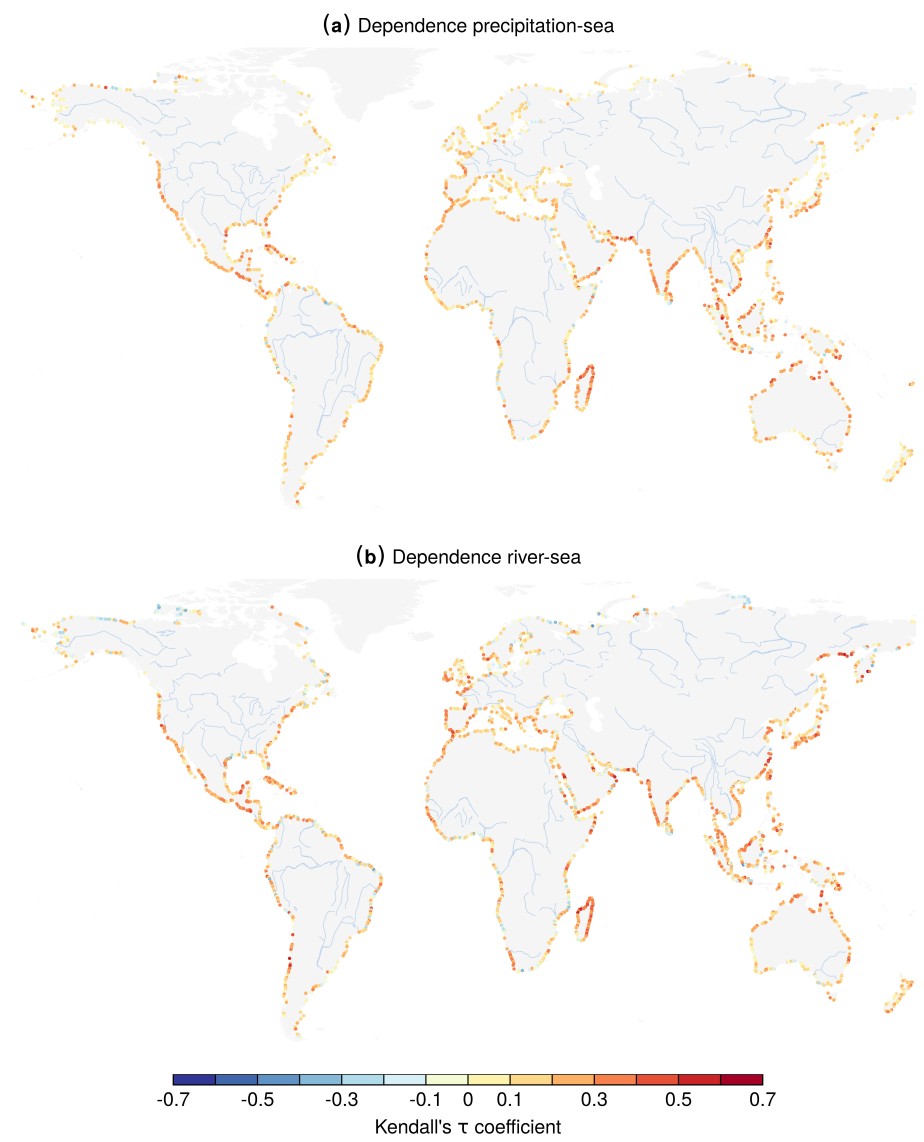

**Figure A1.** Kendall's tau correlation associated with the copulas fitted to the selected pairs of (a) precipitation and sea level, and (b) river discharge and sea level.

## Appendix A: Supporting figures

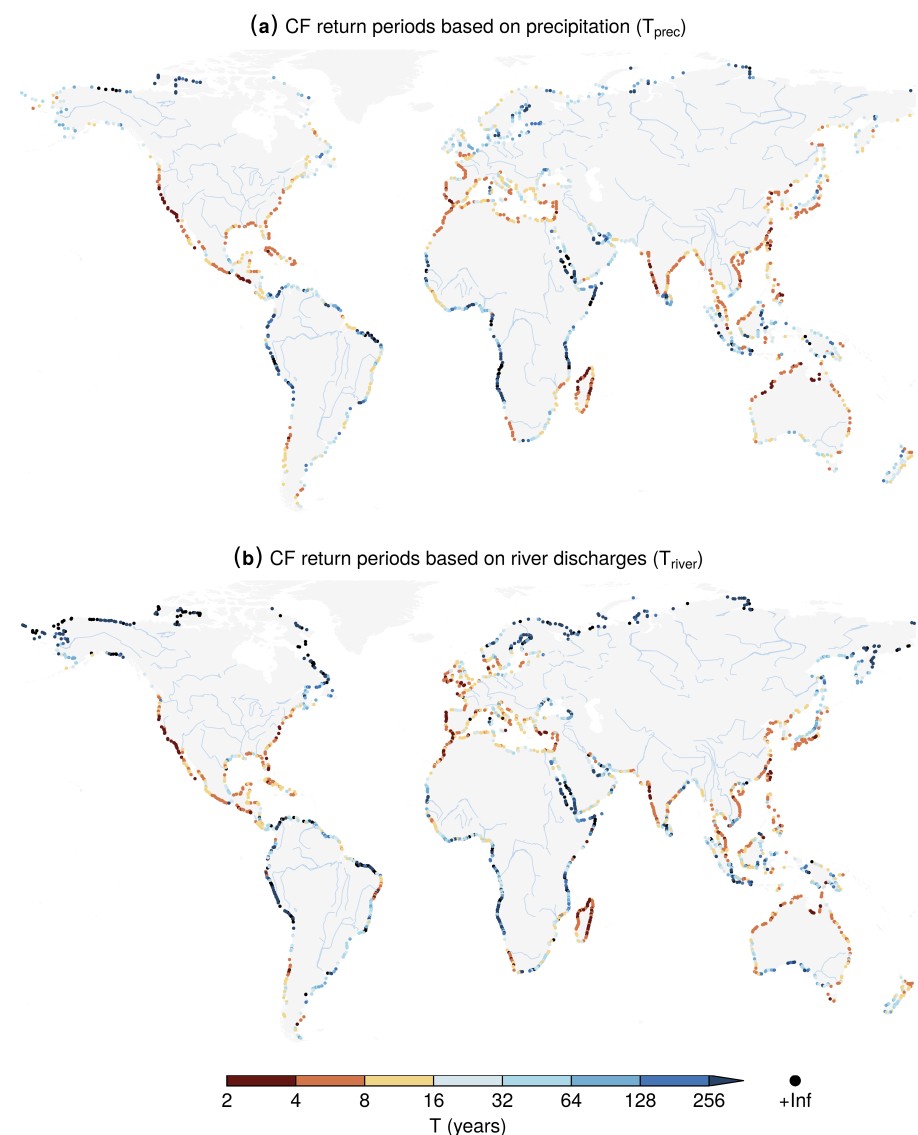

**Figure A2.** The same as Fig. 1, but the extremes are defined as 2-year return levels.

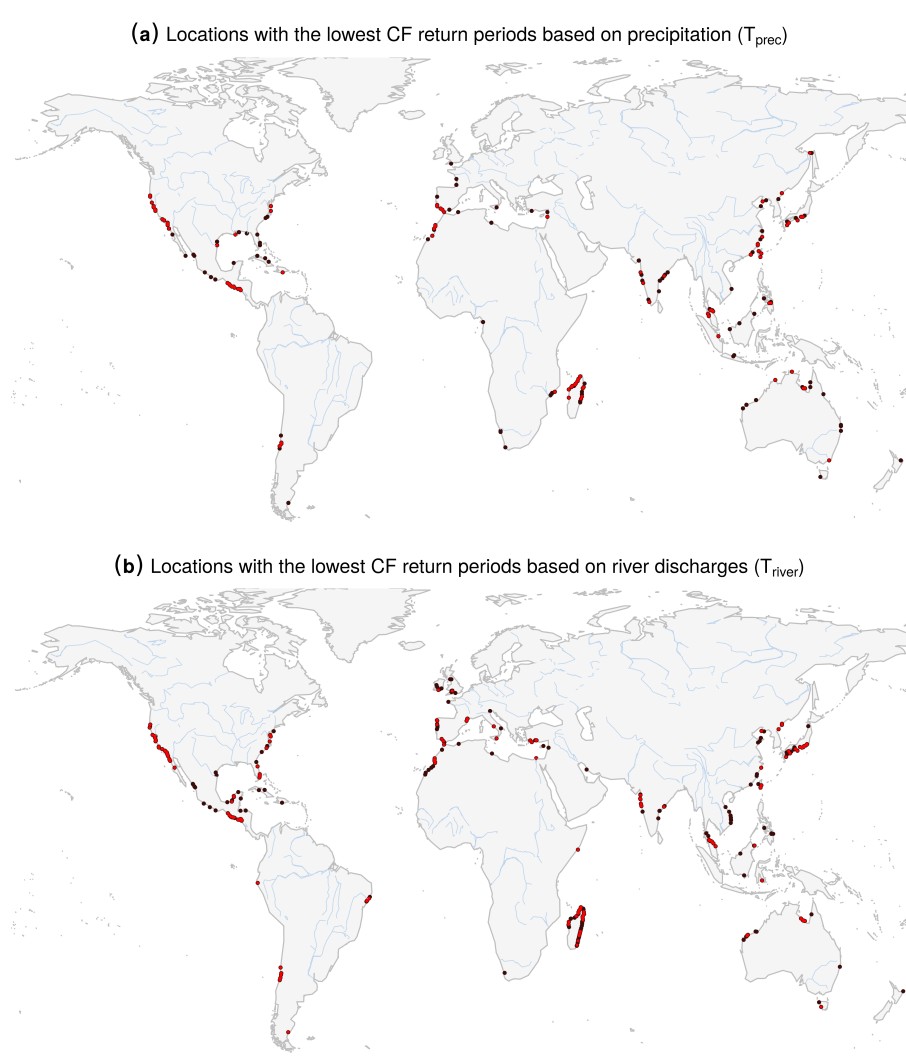

**(a)** Locations with the lowest CF return periods based on precipitation ($T_{prec}$)

**(b)** Locations with the lowest CF return periods based on river discharges ($T_{river}$)

**Figure A3.** Locations with the lowest potential compound flooding probability based on precipitation and on river discharge. Locations with return periods below the 10th and 5th percentile are shown in black and red, respectively. The extremes are defined considering 5-year return levels. CF return periods are based on precipitation in **(a)** and on river discharge in **(b)**.

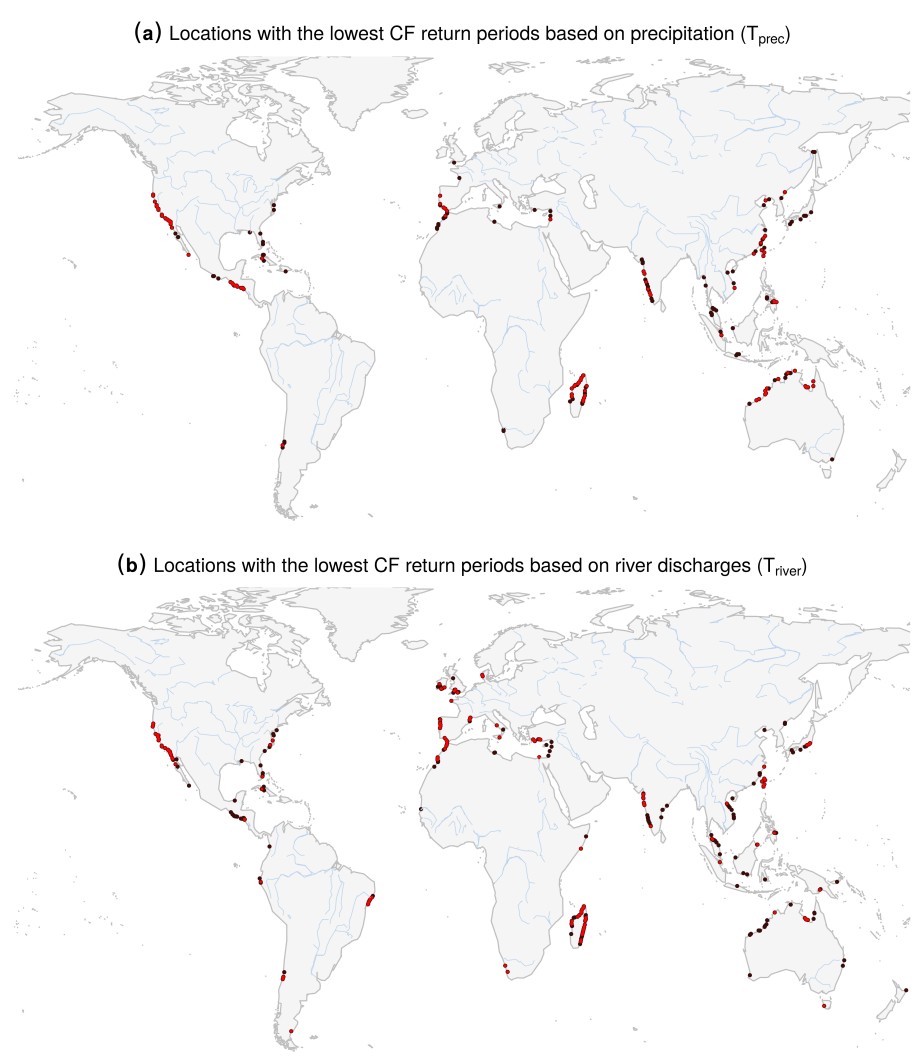

**(a)** Locations with the lowest CF return periods based on precipitation ($T_{prec}$)

**(b)** Locations with the lowest CF return periods based on river discharges ($T_{river}$)

**Figure A4.** The same as Fig. A3, but the results are based on extremes defined considering 2-year return levels.

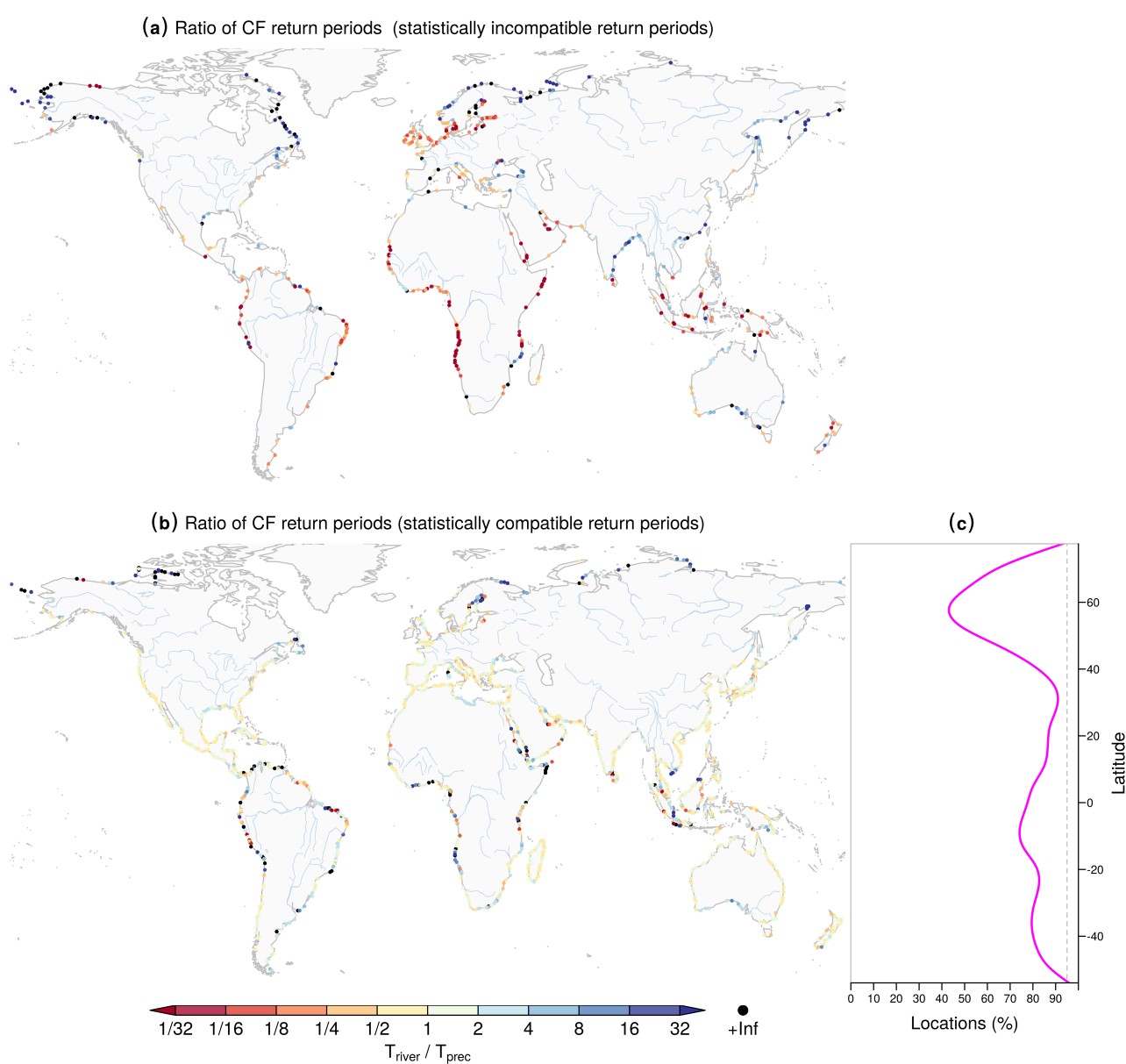

**Figure A5.** The same as Fig. 2, but the results are based on extremes defined considering 2-year return levels.

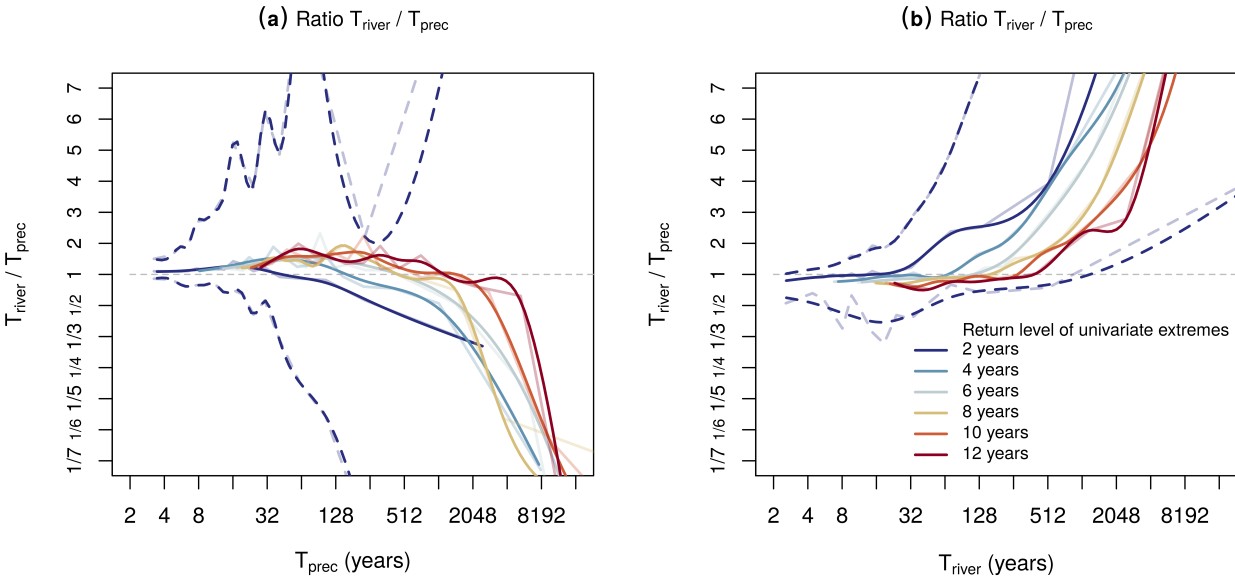

**Figure A6.** (a) Ratio between the CF return periods based on river discharge ($T_{river}$) and on precipitation ($T_{prec}$), as a function of $T_{prec}$ (and of the return levels used to define the CF univariate extremes; see legend in panel (b)). The dashed line is the centered 68% (16th-84th percentiles) confidence interval of the ratio based on the 2-year return levels. Tick lines are obtained through regressing the investigated statistical values to the natural logarithm of the mean return period of the bins via a spline function. (b) The same as (a), but the ratio is conditioned on $T_{river}$. As in Fig. 3d of the original manuscript, a non linear Y-axis for values below 1 is employed such that the specular cases, e.g., ratio r=2 ($T_{river} = 2 \cdot T_{prec}$) and r=1/2 ($T_{prec} = 2 \cdot T_{river}$) (see Fig. 3b) appear symmetric with respect to r=1 ($T_{river} = T_{prec}$).

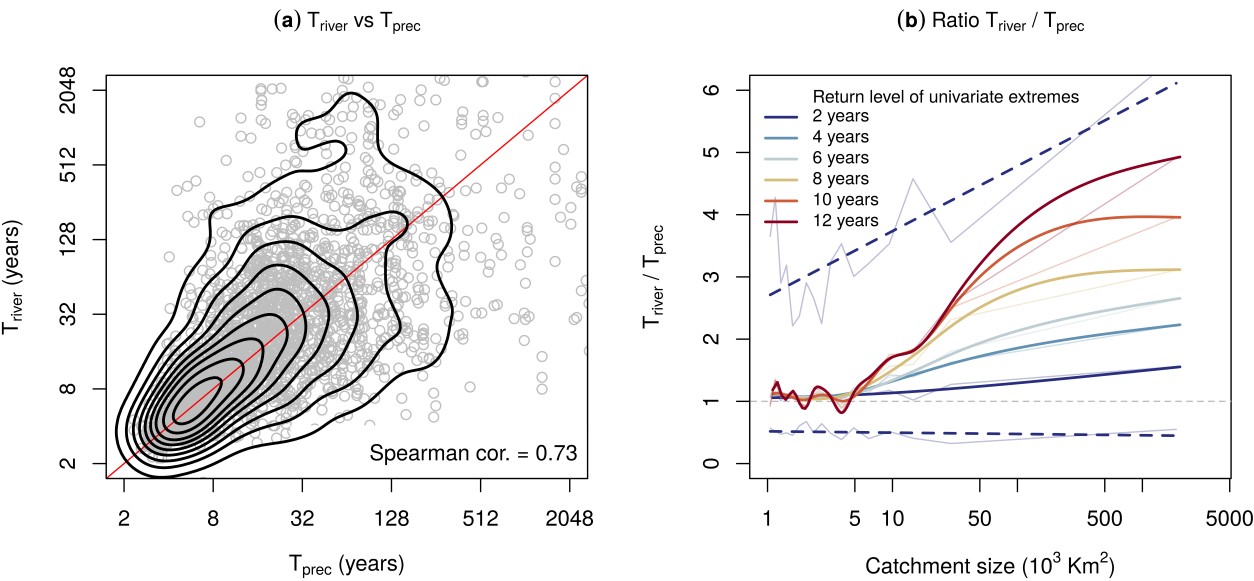

**Figure A7. (a)** The same as Fig. 3a, but the results are based on extremes defined considering 2-year return levels (the figure is based on Fig. A2). **(b)** As Fig. 3d, but with linear y-axis.

*Author contributions.* EB initiated the study, carried out the data analysis, and drafted the manuscript. EB designed the study development with contributions from MIV. EB and MIV worked on the final manuscript version. MIV performed the storm surge runs. All the authors gave conceptual inputs during the data analysis, discussed the results, and commented on the manuscript.

*Competing interests.* The authors declare no competing interests.

*Acknowledgements.* EB acknowledges financial support from the European Research Council grant ACRCC (project 339390). EB acknowledge the European COST Action DAMOCLES (CA17109). MV acknowledges financial support from the EUPHEME project and CoCliServ project, which are part of ERA4CS, an ERA-NET initiated by JPI Climate and co-funded by the European Union (grant no. 690462). The authors would like to thank Lorenzo Mentaschi and Evangelos Voukouvalas who contributed in the generation of the storm surge time series,
and Dirk Eilander for creating and publicly sharing the dataset "Paired time series of daily discharge and storm surge" which was fundamental for developing this study. Finally, the authors would like to thank two anonymous reviewers for their valuable/constructive comments and suggestions that contributed to improving the manuscript.

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
