# Peer review of "Brief communication: The role of using precipitation or river discharge data when assessing global coastal compound flooding"

_Natural Hazards and Earth System Sciences, 2019_

## Referee Comment (RC1) · Anonymous Referee #1 · 6 Feb 2020

This manuscript evaluates the risk of coastal compound flooding at the global scale by using different combinations of drivers (surge vs precip and surge vs discharge) and assess the differences in the results. The analysis is purely based on modelled data. I think the exercise is interesting and provides some important insights that can guide future large-scale assessments of compound flooding risk. The paper is very well written and pleasant to read. I provide some critical comments below which should be addresses before publishing the manuscript. One general comment is that a CF analysis usually starts with assessing dependence between the drivers of interest, in the context of copulas often based on Kendall's rank correlation. This step is completely skipped in the analysis, but a simple comparison of the rank correlation between variable pairs

would already provide useful information, before moving on to the more complex statistical analysis. It would also show where correlation is small/negligible and a complex joint probability analysis possibly not warranted.

L. 94 The threshold of having at least 20 values seems quite low and I suspect the rank correlation that feeds into the copula analysis to be quite sensitive to individual extreme event combinations when the number of pairs is so low. This alone could lead to large differences between the CF estimates for the different variable pairs. It would be interesting to see if there is a relationship between the differences in CF estimates and the data availability, i.e., CF ratio vs sample size. I was also surprised by the short decluster time of only 3 days, given that discharge data is involved in the analysis; the authors actually discuss the aspect of discharge events often lasting longer later on in the manuscript. In many cases discharge can exceed a high threshold for several weeks or even months, and the way the analysis is setup here multiple extreme values are sampled from these events. On the one hand, of course, every time a surge occurs during the high discharge event there could be compound flooding, but on the other hand basic extreme value theory assumptions are violated. I am not saying the entire analysis has to be changed, but would like to hear the authors opinion on this issue, and it might also be worth touching on in the manuscript. What is the lowest threshold that has to be used in order to get to 20 events?

L. 100 Is the same copula always used when the different variable pairs are analyzed (i.e., get the best fit copula for, e.g., surge vs precip and force the same copula to be used for surge vs discharge) or can it change? If the copula is free to change, it might be interesting to test how the results look like wen the same copula is forced (as long as it passes the goodness of fit test(s)).

When displaying the results it would be interesting to see the relationship between CF ratio and absolute CF risk (in relation, for example, to the independence assumption), i.e., are differences between the CF risk relatively larger/smaller in areas where the joint return periods are closer/more different to the independence assumption.

---

## Referee Comment (RC2) · Anonymous Referee #2 · 24 Feb 2020

General Comments:

Bevacqua et al., present a global scale analysis of compound flood (CF) potential by comparing results that estimate the probability of CF using precipitation and storm surge water levels (surge and waves) and discharge and storm surge water levels (surge and waves). This is an important topic for CF research, as a variety of results have been published that use either discharge or precipitation and to my knowledge, this is the first study that tests how the two variables compare. This paper was well written and easy to read. While this is a valuable addition to the literature, I'd like to see the authors more thoroughly address that differences between results are not due

to their analysis techniques before being accepted for publication.

Specific Comments:

The manuscript could address whether results are dependent on the chosen analysis techniques more thoroughly. I'm broadly interested in if the patterns in the return periods and the Triver/Tprec ratio are related to local characteristics of the datasets as the authors suggest in their Results and discussion section, or the analysis that was undertaken, particularly concerning 1) dependency between variables, 2) threshold selection and 3) goodness of fit.

1) Most CF assessments begin by assessing the dependence of the given parameter space. A comparison of the dependence between precipitation and surge level and discharge and surge level may help to explain patterns in the differences in the results. Are there large differences in the dependence between surge vs precip and surge vs river extremes?

2) The authors select pairs of data that are larger than the individual variable's 95 %ile threshold. If less than 20 pairs, the authors decrease the threshold to include more pairs of joint variables. How was the number of pairs, 20, chosen? This seems like a fairly low amount of joint-events to base an analysis on (less than one a year!). Furthermore, how many locations was the threshold decreased, and what's the lowest threshold that was used? Can these variables still be considered "extreme?" It would be interesting to know how variable the number of events analyzed per location was in this analysis.

3) Do patterns in statistical compatibility have anything to do with the goodness of fit of the marginal distributions and copulas? There are some big differences in statistically compatibility and ratio in the same regions (e.g., stations next door to one another). Any reason why that might be?

I think it would be helpful to bring up some more information about the catchments

studied earlier. What's the smallest catchment considered? What's the largest? Are the results dependent on how the catchment values are binned?

Finally, on Page 2, line 53-54, the authors state that the study "aims to assess whether a precipitation based CF assessment can be used as a surrogate for potential CF in estuaries." But I feel like the authors never come back to answering this question. For example, most places that are statistically incompatible are where Tprecip is much larger than Triver. Does this analysis suggest then that precipitation can't be used here, or it should be used in these cases?

Comments on Specific Lines:

Page 1, line 6: In the abstract, the authors state that CF in long river catchments is "more accurately" analyzed using river discharge data. However, there's no underlying assessment of the accuracy of either of these datasets, and/or how well these locations represent joint variables at known locations. Thus the authors may want to consider a change in word choice here.

Technical Corrections:

Page 4, line 108: "centered" is spelled "centred" Page 10, line 199: "Overall, we find that independently of catchment. . ." should be "independent" Figure 3: I don't seem to understand the difference in light versus dark lines in 3d and 3e. All maps could have lat/lon grid lines.

---

## Author Response (AR1)

[revised manuscript text omitted]

When referring to the assessment of whether the CF return periods based on river discharge ($T_{\text{river}}$) are statistically different from those based on precipitation ($T_{\text{prec}}$) or not, we use the concept of statistical compatibility, recently introduced

130 by Amrhein et al. (Amrhein et al., 2019). We compute the  centered 95% (2.5-97.5%) confidence interval of $T_{\text{prec}}$ on the basis of 600 resampled bivariate time series of precipitation and storm surge (each of them built randomly combining observed 1 calendar year bivariate time series (Bevacqua et al., 2020)). $T_{\text{river}}$ is regarded as being statistically compatible with $T_{\text{prec}}$ if $T_{\text{river}}$ lies within the 95% confidence interval of $T_{\text{prec}}$, and incompatible otherwise.

We qualitatively investigate how the two assessments compare for different classes of catchment size. To do so, we rank

135 the rivers based on their catchment size and divide them into groups having the same sample size; for each group we compute different statistics to compare the two assessments: Spearman correlation of $T_{\text{river}}$ and $T_{\text{prec}}$, ratio $T_{\text{river}}/T_{\text{prec}}$, and percentage of locations with $T_{\text{river}}$ compatible to $T_{\text{
[revised manuscript text omitted]
 ($\cancel{T_{prec}}$ $T_{prec}$). Panel $\cancel{(a)}$ **(a)** shows the ratio where $\cancel{T_{river}}$ $T_{river}$ is statistically incompatible (at 95% confidence level, see methods) with $\cancel{T_{prec}}$ $T_{prec}$. Panel $\cancel{(b)}$ **(b)** shows the ratio where $\cancel{T_{river}}$ $T_{river}$ is statistically compatible with $\cancel{T_{prec}}$ $T_{prec}$, while panel $\cancel{(c)}$ **(c)** shows the coastline fraction where this happens (binned every 5° of latitude and smoothed using a spline function). In (c), the dashed grey line shows the 95% level.

[Figure]

**Figure 3.** Comparison of the potential compound flooding probability based on precipitation and on river discharge.  **(a)** Scatterplot (based on Fig. A2) of CF return periods based on river discharge ($\mathcal{T}_{river}$ $T_{\mathrm{river}}$) and on precipitation ($\mathcal{T}_{prec}$ $T_{\mathrm{prec}}$), for extremes defined based on 5-year return levels. Black contours represent the isolines of the kernel density containing from 10% to 90% of the ($\mathcal{T}_{river}, \mathcal{T}_{prec}$ $T_{\mathrm{river}}, T_{\mathrm{prec}}$) pairs. The identity line is shown in red.  **(b)** Empirical probability density function of the variables ($\mathcal{T}_{river}, \mathcal{T}_{prec}$ $T_{\mathrm{river}}, T_{\mathrm{prec}}$) conditioned on the catchment size; the thickness of the isolines increases with the catchment size, such that each bin considers 1/5 of the total number of analysed rivers (3178). Each contour line (isolines of the kernel density) contains about 50% of the ($\mathcal{T}_{river}, \mathcal{T}_{prec}$ $T_{\mathrm{river}}, T_{\mathrm{prec}}$) pairs.  **(c-e)** Statistics comparing $\mathcal{T}_{river}$ $T_{\mathrm{river}}$ and $\mathcal{T}_{prec}$ $T_{\mathrm{prec}}$ as a function of the catchment size and of the return levels used to define the CF univariate extremes (see legend in panel (a)).  **(c)** Spearman correlation between the maps of $\mathcal{T}_{river}$ $T_{\mathrm{river}}$ and $\mathcal{T}_{prec}$ $T_{\mathrm{prec}}$.  **(d)** $T_{\mathrm{river}}/T_{\mathrm{prec}}$ ratio. The dashed line is the centered 68% (16th-84th percentiles) confidence interval of the ratio based on the 2-year return levels. A non linear Y-axis for values below 1 is plotted such that the specular cases, e.g., ratio r=2 ($\mathcal{T}_{river}=2\cdot\mathcal{T}_{prec}$ $T_{\mathrm{river}}=2\cdot T_{\mathrm{prec}}$) and r=1/2 ($\mathcal{T}_{prec}=2\cdot\mathcal{T}_{river}$ $T_{\mathrm{prec}}=2\cdot T_{\mathrm{river}}$) (see Fig. 3b) appear symmetric with respect to r=1 ($\mathcal{T}_{river}=\mathcal{T}_{prec}$ $T_{\mathrm{river}}=T_{\mathrm{prec}}$). (To obtain the plot, the ratio is defined as  $r=T_{\mathrm{river}}/T_{\mathrm{prec}}$; then if r<1, r is transformed to  $r=-T_{\mathrm{prec}}/T_{\mathrm{river}}+2$. Now, e.g., r=2 and r=0 represent specular cases, therefore, where the y-axis is r=0,1,2,..., we can write r=1/2,1/3,1/4,... ). See Fig.  A7b for the plot with standard axis.  **(e)** Percentage of locations where $\mathcal{T}_{river}$ $T_{\mathrm{river}}$ is statistically compatible with $\mathcal{T}_{prec}$ $T_{\mathrm{prec}}$ at the 95% confidence level.  **(f)** Probability density function of the catchment size of the analysed rivers; 50th, 75th, and 95th percentiles of the distribution are shown in red. In  **(c-e)**, tick lines are obtained through regressing the  obtained statistical values per bin (thin lines) to the natural logarithm of the mean size of the catchment bins. All lines are regressed using linear regression, except from the median of the ratio in  **(d)** where a spline function is used. The slopes of the linear regressions are all significant in **(c)**  (p-value < 0.0024), and significant up to the 6-year return level in **(e)**  (p-value < 0.022).

higher return levels, that does not allow for detecting potential differences between large values of $T_{\text{river}}$ and $T_{\text{prec}}$.

The differences between $T_{\text{river}}$ and $T_{\text{prec}}$ are not only controlled by the catchment size, but can be the result of several other factors. During the hydrological modelling, input data artefacts, or model inaccuracies, among others, introduce uncertainty which may contribute to the observed differences. Another important contribution should arise from the uncertainties in the bivariate return period estimation (Bevacqua et al., 2019; Wahl et al., 2015), which can contribute to both positive and negative deviations between $T_{\text{river}}$ and $T_{\text{prec}}$. Moreover, the catchment response time to precipitation depends also on rock and soils catchment permeability (Hendry et al., 2019). Finally, river discharge is influenced not only by local coastal precipitation, but also by the weather over the previous weeks or months over the catchment including evaporation and potentially snowmelt (Couasnon et al., 2020; Bevacqua et al., 2017).

Clearly, the diversity of the physical processes leading to $CF_{\text{prec}}$ and $CF_{\text{river}}$ is very relevant and can cause both positive and negative differences between $T_{\text{river}}$ and $T_{\text{prec}}$ (Blöschl et al., 2019). For example, for any given catchment, a slow catchment response time may either increase or decrease the $T_{\text{river}}/T_{\text{prec}}$ ratio. In locations where cyclones cause frequent concurring storm surge and widespread coastal precipitation, it is not guaranteed that $CF_{\text{river}}$ probability will be also high. For example, if the rainfall in the catchment upstream needs time to reach the coast, long enough for the storm surge to recede, then $CF_{\text{river}}$ will be unlikely and in this case the $T_{\text{river}}/T_{\text{prec}}$ ratio will be high (Klerk et al., 2015; Ward et al., 2018). In contrast, where the $CF_{\text{prec}}$ is unlikely because different weather systems cause precipitation and storm surge extremes, a relatively slow catchment response time may sometimes allow for high river discharge and storm surge to concur if, e.g., a first cyclone causes precipitation driving high river discharge reaching the coast when a second cyclone drives a storm surge (contributing to low $T_{\text{river}}/T_{\text{prec}}$) (Bevacqua et al., 2019).

In addition the presence of a pronounced annual cycle in river discharge can modulate the river discharge driven CF hazard, and thus the $T_{\text{river}}/T_{\text{prec}}$ ratio. In regions where $CF_{\text{prec}}$ is likely, $CF_{\text{river}}$ may be unlikely if the storm surge season does not coincide with the season of the high river discharge (Ward et al., 2018) (high $T_{\text{river}}/T_{\text{prec}}$). In contrast, where the $CF_{\text{prec}}$ is unlikely, $CF_{\text{
[revised manuscript text omitted]

**Answers to comments from anonymous referees 1 and 2**

We would like to thank the reviewers for the time spent in reviewing the paper. We found the comments and suggestions very valuable and constructive. We believe that they contributed to improving the manuscript.

Please, find the response to the individual comments below. Note some tables and figures are located at the end of the file (they are relevant for addressing different comments from both reviewers).

Best regards,
Emanuele Bevacqua (on behalf of all authors)

**Anonymous Referee #1**

*(1) This manuscript evaluates the risk of coastal compound flooding at the global scale by using different combinations of drivers (surge vs precip and surge vs discharge) and assess the differences in the results. The analysis is purely based on modelled data. I think the exercise is interesting and provides some important insights that can guide future large-scale assessments of compound flooding risk. The paper is very well written and pleasant to read. I provide some critical comments below which should be addresses before publishing the manuscript.*

We would like to thank the reviewer for the very positive feedback.

*(2) One general comment is that a CF analysis usually starts with assessing dependence between the drivers of interest, in the context of copulas often based on Kendall's rank correlation. This step is completely skipped in the analysis, but a simple comparison of the rank correlation between variable pairs would already provide useful information, before moving on to the more complex statistical analysis. It would also show where correlation is small/negligible and a complex joint probability analysis possibly not warranted.*

One possibility would be to analyse the dependence between the full distribution of the data, i.e. without focussing on the tails. We think that this might be misleading because, in principle, data might show a poor correlation in the bulk of the distribution but high tail dependence. One could also look at the dependence between the selected "extreme-pairs" used for the copula fitting. Response Fig. 1 shows the dependencies of the pairs used for the fit of the copulas (we show Kendall's tau correlation associated with the copulas fitted to the pairs). A visual inspection of the maps shows that some of the spatial pattern of high dependencies resembles that of low return periods; higher dependencies are found, e.g., along parts of Madagascar and India. However, a low dependence in the tail may also hide a low-return period, if the interarrival time between the selected pairs is low (and vice versa) (see return period expression). Also, the dependence in locations with a different number of selected-pairs may convey different physical information. For these reasons, we have

preferred to directly analyse the CF return periods at all grid points, which in this context provide information on the overall dependence of the pairs (including the behaviour in the tail of the bivariate distribution).

To make things more clear, we have added Response Fig. 1 in the appendix. We refer to this figure in the methods, where we also write: *"Note that, as the return period is obtained as a combination of the average elapsing time \mu and the parametric probability density function of the data in the tail, an exact correspondence between the dependence of the copula and the return period is not expected."* In addition, Response Fig. 1 is also relevant in relation to the next comment of the referee 1 (see below).

[Figure]

**Response Fig. 1.** Kendall's tau correlation associated with the copulas fitted to the selected pairs of (a) precipitation and sea level, and (b) river discharge and sea level.

> *(3) L. 94 The threshold of having at least 20 values seems quite low and I suspect the rank correlation that feeds into the copula analysis to be quite sensitive to*

*individual extreme event combinations when the number of pairs is so low. This alone could lead to large differences between the CF estimates for the different variable pairs. It would be interesting to see if there is a relationship between the differences in CF estimates and the data availability, i.e., CF ratio vs sample size. [...]\* What is the lowest threshold that has to be used in order to get to 20 events?*

*The part of the comment in brackets is addressed later

Thanks for the comment. Employing extreme value theory leads to the advantage of focusing on the tail of the distribution, but also to the disadvantage of having a limited amount of data. In this context, we adopted 20 pairs as a tradeoff between having not too few pairs and having a relatively high threshold. Furthermore, the results are robust with respect to changes in the value N=20. As a first check for the sensitivity of the rank correlation to the number of pairs, please see Response Fig. 1, showing the Kendall's tau of the pairs. The maps indicate that the rank correlations vary pretty smoothly in space, supporting the robustness of the assessment. (Note that, in both maps, some small scale spatial variation in the dependencies may also arise from physical processes. Different catchment characteristics in neighbouring rivers may lead to spatial variations in the river-based dependence; while, to a certain extent, local precipitative events may lead to spatial variations in the precipitation-based dependence.)

Related to "*What is the lowest threshold that has to be used in order to get to 20 events?*", the thresholds used for the fit are relatively high, as shown in Response Fig. 2. See also Response Table 1, showing some specific values of the distribution of the thresholds used in the two analyses, including the minimum values asked explicitly by the referee (based on N=20 pairs). The number N=20 is a compromise between having a large number of pairs, and avoiding obtaining too low thresholds where the values would not be extreme. In fact, to choose this number we had checked the same statistics for N=30 (table 2) and N=40 (table 3). The number N=20 was chosen as it allows for using a threshold above the 90th percentile in at least 90% of locations in both analyses.

[Figure]

**Response Fig. 2.** Thresholds employed for the (a) precipitation and (b) river based assessments.

Based on the suggestion of the referee, we show the CF ratio conditioned on the sample size of the selected pairs (Response Fig. 3). Given that the number of selected pairs depends on the analysis (river- or precipitation-based), we show two plots. No clear relationship between the median ratio Triver/Tprec (thick lines) and the sample size appears from Fig. 3. However, there is a weak tendency towards slightly higher ratios for smaller sample size, therefore we did carry some additional tests to make sure that the results are not affected by a low number in selected pairs.

We carried out several tests during the development of the study. These were important to support the choice of N=20 pairs and are relevant in this context also to answer the concerns of the reviewer regarding the effect of a potentially too low value N. The tests highlighted that the return periods are robust, i.e. insensitive to reasonable changes in this value N. This is shown in the Response Fig. 8 at the end of the file for return periods associated with return levels of 2 and 5 years, and for N=20 (used in the manuscript) and N=40. Therefore, no differences in the two assessments are expected because of the choice of N=20. Given the robustness of the results for changes in N, we did choose N=20 that, as discussed above, is

a compromise between having a large number of pairs and avoiding obtaining too low thresholds where the selected values would not be extreme.

We now discuss the topic of the thresholds explicitly in the paper, which is actually very important to avoid any potential concerns to the reader:

*"If the defined thresholds result in a small group of selected pairs, we lower the 95th percentile selection threshold to guarantee having at least 20 pairs. The choice of 20 pairs is a tradeoff between having a sufficient amount of selected pairs and employing a reasonably high threshold for the fit of the parametric distribution in the tail. Furthermore, the return periods are largely insensitive to reasonable changes in the threshold (results are similar based on 20, 30, and 40 pairs; not shown). The selection thresholds are generally high: 75% of the locations have a selected-threshold larger or equal to 0.95 and 0.94 for the precipitation- and river-based analysis, respectively. And 95% (99%) of the locations have a selected-threshold above 0.93 (0.885) and 0.89 (0.85) for the precipitation- and river-based analysis, respectively "*

[Figure]

**Response Fig. 3.** Ratio Triv/Tprec against the number of selected pairs. (a) Ratio between the CF return periods based on river discharge (Triver) and on precipitation (Tprec), as a function of the number of selected pairs employed to fit the bivariate parametric distribution for the precipitation based assessment (and of the return levels used to define the CF univariate extremes; see legend in panel (a)). (b) The same as (a), but the ratio is conditioned on the number of selected pairs employed to fit the bivariate parametric distribution for the river based assessment. To obtain the plot, locations are ranked based on the number of selected pairs, and then divided into groups, each containing the same number of locations. For each group, we then compute and show the median ratio. (Note that we employ the same range of the y-axis (1/6, 6) as that of Fig. 3d in the original manuscript, which allows for direct comparison of the figures).

*(4) I was also surprised by the short decluster time of only 3 days, given that discharge data is involved in the analysis; the authors actually discuss the aspect of discharge events often lasting longer later on in the manuscript. In many cases discharge can exceed a high threshold for several weeks or even months, and the way the analysis is setup here multiple extreme values are sampled from these events. On the one hand, of course, every time a surge occurs during the high discharge event there could be compound flooding, but on the other hand basic extreme value theory assumptions are violated. I am not saying the entire analysis has to be changed, but would like to hear the authors opinion on this issue, and it might also be worth touching on in the manuscript.*

We agree with the referee that "In many cases discharge can exceed a high threshold for several weeks or even months, and the way the analysis is set up here multiple extreme values are sampled from these events." and therefore we agree that this is a point deserving an explicit discussion within the manuscript. The reason for setting up the analysis in this way is exactly that mentioned by the referee, i.e. "every time a surge occurs during the high discharge event there could be compound flooding". Therefore, considering only one of the several storm surges that occur during a period of high river flow would lead, in practice, to an underestimation of the CF potential. (This is clear when thinking, e.g., about New Orleans (Louisiana, US) that in 2019 experienced extremely high water discharge from the Mississippi River from March to July. Any storm surge in that period could have led to compound flood potential, hence they all should be taken into account in an assessment of the hazard.) Therefore, we believe that it is important to follow a pragmatic approach and adopt this choice of considering all the storm surge events. We discuss this topic in the Methods when introducing the declustering, where we add:

*"While this choice has the drawback of not fully respecting the assumptions of independent realisations of the extreme events, which is necessary to apply extreme values theory in its generic form, it allows considering multiple storm surges that may occur during a sustained period of high river discharge and that could lead to multiple compound floods."*

*(5) L. 100 Is the same copula always used when the different variable pairs are analyzed (i.e., get the best fit copula for, e.g., surge vs precip and force the same copula to be used for surge vs discharge) or can it change? If the copula is free to change, it might be interesting to test how the results look like wen the same copula is forced (as long as it passes the goodness of fit test(s)).*

First, we observe that the dependence between the variables is captured within the analysis both by the interarrival time between select pairs (that is smaller for higher-dependent data) and by the copula itself. Both, naturally, contribute to the differences between the two assessments.

Regarding the specific question, we observe that for any given location the physical processes captured by the two assessments can differ, therefore there is no physical justification for employing the same copula. In particular, it should be also considered that

there are no available tools for defining which is the "right" copula structure and we do not have prior physical knowledge of the true structure of dependence at the different locations. Therefore, we have employed the same criterion for selecting the copula in both individual assessments and allowed the copulas to be different in the two assessments. (We have of course applied statistical tests to investigate whether the fitted copulas could be rejected or not, and the results indicate that the chosen copulas are consistent with the data.) For the reasons above, we think that an interpretation of the proposed analysis would not be straightforward.

Based on the question of the referee, we have added a sentence in the manuscript where we state explicitly, to avoid misunderstandings, that different copulas are allowed at the same location: "*In general, the physical processes captured by the two assessments can differ (even at the same location), therefore we allow for the selection of different copulas in the two assessments.*"

> *(6) When displaying the results it would be interesting to see the relationship between CF ratio and absolute CF risk (in relation, for example, to the independence assumption), i.e., are differences between the CF risk relatively larger/smaller in areas where the joint return periods are closer/more different to the independence assumption.*

We think that investigating whether the differences between the two return periods is more pronounced for high return periods (of Tprec or Triver) is a very valuable suggestion. The relationship between the CF ratio and the absolute CF return periods is somehow visible from Fig. 1 and 2, however, we had not considered this explicitly. We show the relationship in Response Fig. 4 (added to the new version of the manuscript). The figure reveals that there is a tendency towards higher differences in the two assessments in locations where either or both Tprec and Triver are high. The differences tend to be small at locations with small return periods.

We explain the behaviour seen in Response Fig. 4a based on the following reasoning (similar reasoning applies for Response Fig. 4b). First it should be considered that (1) Tprec and Triver are correlated (Fig.1 and Fig. 3a) and that (2) large return periods are highly uncertain.
The combination of (1) and (2) provides insight on why:
- the ratio Triver/Tprec conditioned to locations characterized by low Tprec is about equal to 1 (Response Fig. 4a). Locations where Tprec is low are characterized, on average, by a low Triver (because of 1). Low return periods have relatively low uncertainties, therefore the selected Triver (associated with low Tprec) is on average similar to Tprec. This ultimately leads to a ratio about equal to 1.
- Similarly, the ratio Triver/Tprec conditioned to locations characterized by high Tprec is low (Response Fig. 4a). Locations where Tprec is high are characterized, on average, by high Triver (because of 1). However, while Tprec is very high and fixed due to the conditioning itself, the selected values of Triver are largely variable because of (2). As a result, the average value of Triver is smaller than the Tprec (causing low ratio).

[Figure]

**Response Fig. 4.** (a) Ratio between the CF return periods based on river discharge ($T_{river}$) and on precipitation ($T_{prec}$), as a function of $T_{prec}$ (and of the return levels used to define the CF univariate extremes; see legend in panel (b)). The dashed line is the centered 68\% (16th-84th percentiles) confidence interval of the ratio based on the 2-year return levels. Tick lines are obtained through regressing the investigated statistical values to the natural logarithm of the mean return period of the bins via a spline function. (b) The same as (a), but the ratio is conditioned on $T_{river}$. As in Fig. 3d of the original manuscript, a non linear Y-axis for values below 1 is employed such that the specular cases, e.g., ratio r=2 ($T_{river}=2 \cdot T_{prec}$) and r=1/2 ($T_{prec}=2 \cdot T_{river}$) (see Fig. 3b) appear symmetric with respect to r=1 ($T_{river}=T_{prec}$).

Fig. Response Fig. 4 provides insights also on whether the "differences between the CF risk is relatively larger/smaller in areas where the joint return periods are closer/more different to the independence assumption". In fact, the differences between the CF assessments (the divergence from 1) tend to occur - on average - at higher return periods (x-axis) for the assessment based on higher extreme values (colors towards the red). That is, the ratio diverges from 1 while the return period approaches values consistent with independent drivers (that are higher for return periods associated with extremes defined as higher return levels).

Overall, we believe that the suggested investigation provides also some general and interesting insights of practical interest. In fact, it implies that practically the discrepancies between the approaches are large, especially where the actual CF potential tends to be low (where CF drivers approach independence), i.e. where an analysis of compound flooding is less relevant. Hence, the precipitation-based analysis is similar to the river-based where it is actually more relevant to analyse the CF hazard, which strengthens some of the conclusions of the work.

When presenting the results, we write: *"We find that there is a tendency towards higher differences in the two assessments at locations where either or both Tprec and Triver are*

*high (i.e. where T approaches the value expected under independence of the CF drivers) (Fig. A6). (This appears consistent with the high uncertainty associated with large CF return periods.) Such a finding has relevant implications, as it indicates that the two assessments tend to be similar, on average, where assessing the actual CF is more important, i.e. where there is a relatively high CF potential (Fig. A6)."*

In the abstract and conclusions, we added: *"on average, the deviations between the two assessments are smaller in regions where assessing the actual CF is more relevant, i.e. where the CF potential is high."*

**Anonymous Referee #2**

*Bevacqua et al., present a global scale analysis of compound flood (CF) potential by comparing results that estimate the probability of CF using precipitation and storm surge water levels (surge and waves) and discharge and storm surge water levels (surge and waves). This is an important topic for CF research, as a variety of results have been published that use either discharge or precipitation and to my knowledge, this is the first study that tests how the two variables compare. This paper was well written and easy to read. While this is a valuable addition to the literature, I'd like to see the authors more thoroughly address that differences between results are not due to their analysis techniques before being accepted for publication.*

We would like to thank the reviewer for the very positive feedback, highlighting the timeliness of our work.

*Specific Comments: The manuscript could address whether results are dependent on the chosen analysis techniques more thoroughly. I'm broadly interested in if the patterns in the return periods and the Triver/Tprec ratio are related to local characteristics of the datasets as the authors suggest in their Results and discussion section, or the analysis that was undertaken, particularly concerning 1) dependency between variables, 2) threshold selection and 3) goodness of fit.*

*1) Most CF assessments begin by assessing the dependence of the given parameter space. A comparison of the dependence between precipitation and surge level and discharge and surge level may help to explain patterns in the differences in the results. Are there large differences in the dependence between surge vs precip and surge vs river extremes?*

Given that the extremes are defined based on return level values, the present analysis of potential compound flooding is directly linked to the dependence of the variables. Therefore, yes, the results are well explained by differences in the dependence. We now add a sentence in the methods: *"Overall, given the definition of the extremes based on $\alpha$-year return levels, the bivariate return period is inherently linked to the dependence of the pairs in the tail of the distribution."*

Given that this question is similar to that asked by referee 1, please see the answer to comment (2) of referee 1 above.

*2) The authors select pairs of data that are larger than the individual variable's 95 %ile threshold. If less than 20 pairs, the authors decrease the threshold to include more pairs of joint variables. How was the number of pairs, 20, chosen? This seems like a fairly low amount of joint-events to base an analysis on (less than one a year!). Furthermore, how many locations was the threshold decreased, and what's the lowest threshold that was used? Can these variables still be considered "extreme?" It would be interesting to know how variable the number of events analyzed per location was in this analysis.*

First of all, it could be relevant to highlight that all the variables are used to estimate the return period, not only the selected pairs. As shown in equation 1 in the main text, the return period is a function of the interarrival time between selected pairs and the parametric probability density function modelling the selected pairs. That is, this is a semi-parametric return period (line 90 of the original manuscript), as it is always the case for the corresponding univariate case of return periods based on peaks over threshold.

We agree that the number of N=20 pairs is somewhat arbitrary. However, it was chosen after carrying some tests (both during this and previous studies) for the bivariate return periods. About the variability of the number of selected pairs, there is a tendency towards having fewer pairs at locations where CF is unlikely. This is consistent with the fact that there are fewer concurrences of extremes there. The median number of selected pairs for the river-based analysis is 30 (interquartile range (22,57)), and for the precipitation-based analysis is 58 (interquartile range (30,86)).

The tests that we carried out to choose N=20 as threshold showed that the return periods are robust with respect to reasonable changes in this value N. This is shown in the Response Fig. 8 at the end of the file for return periods associated with return levels of 2 and 5 years, and for N=20 (as in the manuscript) and N=40.

Furthermore, to answer other specific questions, please see Table 1 at the end of the file. It shows some specific values of the distribution of the thresholds used in the two analyses (based on N=20 pairs). The number N=20 is a compromise between having a large number of pairs, and avoiding obtaining too low thresholds whereas the values would not be extreme anymore. In fact, to choose this number we had checked the same statistics for N=30 (table 2) and N=40 (table 3). The number N=20 was chosen as it allows for using a threshold above the 90th percentile in at least 90% of locations in both analyses. (For example, using N=40, the lowest threshold would be 0.68, i.e. we would not be in the tail of the distribution anymore.) This is important to stay within the framework of extreme values, as also the referee points out asking "Can these variables still be considered "extreme?". Hence, based on the way we chose N, the variables can indeed be reasonably considered as extremes.

To explicitly address this topic, we have extended the method section:

*"If the defined thresholds result in a small group of selected pairs, we lower the 95th percentile selection threshold to guarantee having at least 20 pairs. The choice of 20 pairs is a tradeoff between having a sufficient amount of selected pairs and employing a reasonably high threshold for the fit of the parametric distribution in the tail. Furthermore, the return periods are largely insensitive to reasonable changes in the threshold (results are similar based on 20, 30, and 40 pairs; not shown). The selection thresholds are generally high: 75% of the locations have a selected-threshold larger or equal to 0.95 and 0.94 for the precipitation- and river-based analysis, respectively. And 95% (99%) of the locations have a selected-threshold above 0.93 (0.885) and 0.89 (0.85) for the precipitation- and river-based analysis, respectively "*

*3) Do patterns in statistical compatibility have anything to do with the goodness of fit of the marginal distributions and copulas? There are some big differences in statistically compatibility and ratio in the same regions (e.g., stations next door to one another). Any reason why that might be?*

In the following, the main concepts are highlighted in bold.

As stated in the methods, the goodness of fit of the marginal distributions and of the copulas were tested, i.e. three distributions (marginals and copula) per assessment were tested (for a total of six distributions tested per location given the two assessments). For a given distribution, i.e. a marginal or a copula distribution, less than about 5% (depending on the distribution) of the p-values are below 0.05, which is an acceptable range as 5% is the value expected under the hypothesis that the data are distributed according to the fitted distribution (Zscheischler et al., 2017). (Note that there is no evident spatial pattern of the locations with p-values below 0.05.)

Furthermore, **we did not find a relationship between the p-values and the statistics/results**. This is shown in *Response Fig. 5*, which is the same as Fig. 3c-d of the manuscript, but obtained as it follows. The first row shows the panel obtained when focussing the analysis only on locations where all the distributions cannot be rejected at 95% confidence level; the second focuses on locations where at least one distribution can be rejected. Naturally, as the latter is based on a lower number of locations, the associated results are more noisy. However, the main conclusions obtained from the two analyses are the same. (Because of this result and of the reasoning of the previous paragraph, we have considered all the locations in the analysis.) Similarly, we also visually inspected (not shown) the map of the ratio Triver/Tprec based only (i) on locations where all the distributions cannot be rejected at 95% confidence level and (ii) only locations where at least one distribution can be rejected. Consistently with the above, in both maps neighbor locations can have large differences in the ratio (ratios>>1 and ratio<<1). Therefore, goodness of fit seems to not be related to the large differences seen in neighborn locations.

[Figure]

*Response Fig. 5. As Fig. 3c-e of the original manuscript, but the first line shows the results obtained for locations where all the distributions involved in the return period computation "pass" the test, and in the second line the location where at least one distribution does not "pass" the test (see text). In panel b, we show the obtained statistical values per bin with the thick line. In a and c, as in the paper, thick lines are obtained through linearly regressing the obtained statistical values per bin (shown with thin lines) to the natural logarithm of the mean size of the catchment bins.*

**Some of these differences (in the map of the ratio Triver/Tprec) between neighboring locations may be explained by differences in basin characteristics. However, we think that the main source of these differences could be return period uncertainties.** Indeed, neighborn locations having large differences in the ratio appear frequently in locations where return periods tend to be high, for example in the tropics (Fig. 2). Large return periods are subject to large uncertainties both because of sampling uncertainty and because of model uncertainties (e.g. bivariate model fitting and definition of the return levels). This is indicated by Response Fig. 6a where the 95% confidence interval of Tprec at different locations is shown as a function of the return period Tprec. As a result of the high uncertainty associated with high return periods, in areas where CF is unlikely (e.g., in small catchments of the tropics), return periods are particularly uncertain, hence the estimated return periods can be substantially different (positively or negatively) from the expected one (i.e., the "real" one) (this is indicated by Response Fig. 6b-c). As a result, in locations with high return periods, even assuming that the expected return period from the two assessments being similar, substantial discrepancies in the two assessments (leading to high (>>1) or low (<<1) ratios) may arise from return period uncertainties.

This effect provides a plausible explanation, from a statistical point of view, of why it is possible to find ratio>>1 and ratio<<1 in neighborn locations. The effect can also explain differences in compatibility in nighborn locations: these are consistent with the effect itself combined with the fact that the actual expected return periods from the two assessments can be different enough to lead to differences that are not only large but also significant.

We have added a paragraph in the paper: "In areas with a tendency towards high CF return periods, e.g. the tropics, neighbour locations show divergent values in the ratio between the return periods of the two assessments (dark blue and red dot in Fig. 2). Further tests showed that this behaviour is not related to goodness of fit of the bivariate distributions (see discussion in the response to the reviewer 1), rather it appears associated with the large uncertainties of high return periods and potentially with different catchment characteristics."

[Figure]

**Response Fig. 6.** (a) Amplitude of the centered 95% confidence interval of T (i.e., difference between the percentiles: T_97.5th - T_2.5th) as a function of the T. Each dot is a different location, and the figures are obtained based on the precipitation based assessment (similar results would be obtained based on river discharge). Panel b and c indicate that very large (>>1) or small (<<1) values of the ratio between the estimated T and the actual T can occur for large T as a result of uncertainties. (b) Ratio $T_{97.5} / T$ as a function of T. (c) Ratio T_2.5 / T as a function of T. In (a) and (b), green dots mark locations for which the value on the y-axis is infinite. The 95% confidence interval of T is obtained based on resampling procedure (see Methods).

> *I think it would be helpful to bring up some more information about the catchments studied earlier. What's the smallest catchment considered? What's the largest? Are the results dependent on how the catchment values are binned?*

We agree. We have now added information on the catchment at the beginning and end of the data section:
- *"We consider river discharge daily maxima from a publicly available global dataset (Eilander, 60 2019; Couasnon et al., 2019), which includes coastal catchments larger than 1,000 km$^{2}$."*
- *"We analyse CF only around river mouth locations whose nearest precipitation and storm surge grid points lie within a distance of 75 km (Couasnon et al., 2019). This results in considering locations at river mouths of catchments with size in between*

*about 1,000 and 3,690,000 Km$^{2}$ (95\% having size smaller than 50,000 Km$^{2}$; Fig. 3f)."*

About the binning: Overall, the results of the binning procedure are meant to provide qualitative information, as we also state in the paper: "We qualitatively investigate how the two assessments compare for different classes of catchment size." (line 112), and adopted this binning procedure as it "provides equally robust statistics for each bin" (line 115).

In the manuscript we state that "we rank the rivers based on their catchment size and divide them into groups having the same sample size; for each group we compute different statistics to compare the two assessments". It is possible to bin rivers in larger or smaller numbers of groups, corresponding to less or more populated groups, respectively. A too less populated group would lead to more noisy results, on the contrary a too small number of groups (more populated) would not allow for satisfactorily exploring the variability of the statistics as a function of the catchment size. The chosen number of groups is therefore a trade off between these two situations above. **While the results are slightly different when changing the number of groups, the conclusions are unchanged (as mentioned in the original version of the manuscript), e.g., see Response Fig. 7. In the paper, we employ N=17.**

[Figure]

**Response Fig. 7.** *As Fig. 3c-e of the original manuscript, but employing a different number of bins (shown at the bottom-right of each panel).* Compared to the images of the paper, we do show only the rough statistics (without regressing any spline or line to them, which allows for better appreciating the differences between the graphs).

*Finally, on Page 2, line 53-54, the authors state that the study "aims to assess whether a precipitation based CF assessment can be used as a surrogate for potential CF in estuaries." But I feel like the authors never come back to answering this question. For example, most places that are statistically incompatible are where Tprecip is much larger than Triver. Does this analysis suggest then that precipitation can't be used here, or it should be used in these cases?*

We thank the reviewer for this comment. In locations where the two return periods are incompatible, one should not employ precipitation for assessing the river based return period. The main aim of the study is not to address whether the two return assessments are interchangeable at the specific locations, rather when carrying out large-scale assessments. However, we agree that also indications for local assessments can be provided.

Although the discussion in the introduction was based on the large-scale CF assessment, we see that in the specific text quoted by the referee, and more specifically in the final paragraph of the introduction, there is no reference to the "large-scale" nature of the assessment. This could indeed be misleading for the reader. Therefore, we have now added some words/sentences in the text (shown in below). In the introduction:

*"Given the scarcity and heterogeneous distribution of in situ data (Ward et al., 2018; Couasnon et al., 2019; Wu et al., 2018), scientists have started to employ model data - of river, storm surge, and precipitation - to assess the **large-scale** potential CF hazard (Ward et al., 2018; Bevacqua et al., 2019a; Wu et al., 2018; Couasnon et al., 2019; Wu et al., 2018; Paprotny et al., 2018; Bevacqua et al., 2019b). Against the foregoing background, the present study aims to assess whether a precipitation based **large-scale** CF assessment can be used as a surrogate for potential CF in estuaries at the **large-scale**. To that end we use coherent global model datasets of storms surges (including wave effects) (Vousdoukas et al., 2018), precipitation (Beck et al., 2017b), and river discharge (Couasnon et al., 2019; Eilander, 2019) and conduct a first global comparison of the results obtained through the two approaches, keeping all the other methodological aspects identical."*

Within this context, we feel like we address the question related to assessing "whether a precipitation based **large-scale** CF assessment can be used as a surrogate for potential CF in estuaries at the **large-scale**". In the conclusions, we explicitly refer to the large-scale nature of the assessment and based on the analyses presented, we state: "This study indicates that for these large-scale assessments, a precipitation-based CF analysis can provide satisfactory information on the CF potential in estuaries of small and medium size rivers (catchment smaller than about 5-10,000 Km2 )."

Furthermore, to even more explicitly address the comment of the referee about specific locations, we have also modified the one of the last sentences of the paper:

*"Naturally, employing river discharge data should always be preferred to using precipitation when studying both the large- and local-scale CF in estuaries, when data are available,*

**especially in areas where we detected large differences between the two approaches**. *The importance of using river discharge data is even greater in estuaries of long rivers."*

> *Comments on Specific Lines:*

> *Page 1, line 6: In the abstract, the authors state that CF in long river catchments is "more accurately" analyzed using river discharge data. However, there's no underlying assessment of the accuracy of either of these datasets, and/or how well these locations represent joint variables at known locations. Thus the authors may want to consider a change in word choice here.*

We changed the text: "...CF in long rivers (catchment $\gtrsim$ 5-10,000 Km$^2$) should be analysed using river discharge data."

> *Technical Corrections:*

> *Page 4, line 108: "centered" is spelled "centred"*
> *Page 10, line 199: "Overall, we find that independently of catchment. . ." should be "independent"*
> *Figure 3: I don't seem to understand the difference in light versus dark lines in 3d and 3e.*
> *All maps could have lat/lon grid lines.*

Thanks, we have adopted these changes, apart from the lat/lon grid lines that - in our opinion - are not crucial in the context of the paper. However, we would be certainly happy to add them at a later stage if also the editor shares the same opinion as the reviewer.

**References**

*Zscheischler J, Seneviratne SI. Dependence of drivers affects risks associated with compound events. Science advances. 2017 Jun 1;3(6):e1700263.*

[Figure]

**Response Fig. 8.** Comparison between the assessments based on a different number of minimum required selected pairs for the fit of the bivariate parametric distribution. Top 4 panels based on N=40 pairs; Bottom 4 panels based on N=20 pairs. We show the return periods based on extremes defined as 2- and 5-year return levels (as in the manuscript) on the left and right column, respectively.

| Nmin=20 | | |
| --- | --- | --- |
| | Selected-threshold of the precipitation-based analysis | Selected-threshold of the river-based analysis |
| Minimum value | 0.785 | 0.775 |
| 1st percentile | 0.885 | 0.84 |
| 10th percentile | 0.935 | 0.91 |
| 25th percentile | 0.95 | 0.94 |
| 50th percentile | 0.95 | 0.95 |

Response Table 1

| Nmin=30 | | |
| --- | --- | --- |
| | Selected-threshold of the precipitation-based analysis | Selected-threshold of the river-based analysis |
| Minimum value | 0.765 | 0.745 |
| 1st percentile | 0.86385 | 0.815 |
| 10th percentile | 0.92 | 0.895 |
| 25th percentile | 0.95 | 0.925 |
| 50th percentile | 0.95 | 0.95 |

Response Table 2

| Nmin=40 | | |
| --- | --- | --- |
| | Selected-threshold of the precipitation-based analysis | Selected-threshold of the river-based analysis |
| Minimum value | 0.755 | 0.68 |
| 1st percentile | 0.845 | 0.795 |
| 10th percentile | 0.905 | 0.88 |
| 25th percentile | 0.935 | 0.91 |
| 50th percentile | 0.95 | 0.935 |

Response Table 3